# Clustering in Deep Stochastic Transformers

Lev Fedorov [1]   Michaël E. Sander [2]   Romuald Elie [2]   Pierre Marion [3]   Mathieu Laurière [1]

## Abstract

Transformers have revolutionized deep learning across various domains but understanding the precise token dynamics remains a theoretical challenge. Existing theories of deep Transformers with layer normalization typically predict that tokens cluster to a single point; however, these results rely on deterministic weight assumptions, which fail to capture the standard initialization scheme in Transformers. In this work, we show that accounting for the intrinsic stochasticity of random initialization alters this picture. More precisely, we analyze deep Transformers where noise arises from the random initialization of value matrices. Under diffusion scaling and token-wise RMS normalization, we prove that, as the number of Transformer layers goes to infinity, the discrete token dynamics converge to an interacting-particle system on the sphere where tokens are driven by a *common* matrix-valued Brownian noise. In this limit, we show that initialization noise prevents the collapse to a single cluster predicted by deterministic models. For two tokens, we prove a phase transition governed by the interaction strength and the token dimension: unlike deterministic attention flows, antipodal configurations become attracting with positive probability. Numerical experiments confirm the predicted transition, reveal that antipodal formations persist for more than two tokens, and demonstrate that suppressing the intrinsic noise degrades accuracy.

## 1. Introduction

The Transformer architecture (Vaswani et al., 2017) has revolutionized modern machine learning, establishing itself as the backbone of state-of-the-art models across distinct modalities (Devlin et al., 2019; Dosovitskiy et al., 2021;

[1]New York University [2]Google DeepMind [3]Inria, École Normale Supérieure, PSL Research University. Correspondence to: Lev Fedorov <lev.fedorov@nyu.edu>.

*Proceedings of the 43rd International Conference on Machine Learning*, Seoul, South Korea. PMLR 306, 2026. Copyright 2026 by the author(s).

Moussad et al., 2023). Despite this empirical dominance, a rigorous understanding of the signal propagation in deep, randomly initialized Transformers remains limited. The attention mechanism (Bahdanau et al., 2014) introduces a global, nonlinear interaction whose asymptotic behavior in deep networks is understood only in fragments.

Among existing theoretical directions, mean-field and infinite depth limits of self-attention dynamics (Sander et al., 2022) have yielded a rich framework, explaining the emergence of clustering (Geshkovski et al., 2023; Criscitiello et al., 2024), rank collapse (Noci et al., 2022; Giorlandino & Goldt, 2025), metastability (Bruno et al., 2024), and geometric structures that mirror empirical behavior (Castin et al., 2025; Burger et al., 2025). However, these works typically rely on idealized structural assumptions: they consider weights that are tied across the layers, or equivalently assume a fully deterministic layer dynamics, thereby neglecting the layer-wise randomness inherent to initialization.

In parallel, several theoretical studies have incorporated noise into attention models (Shalova & Schlichting, 2024; Balasubramanian et al., 2025). While these works capture qualitative phenomena such as phase transitions and metastability of token configurations, the injected noise is typically external and *idiosyncratic*, acting independently at the token level. Consequently, these models lack a self-contained derivation from the forward pass of implementable Transformer architectures and do not capture the *common-noise* structure induced by layer-wise random initialization.

A third relevant direction is diffusion limits for residual networks (Peluchetti & Favaro, 2020; 2021; Cont et al., 2023; Li et al., 2022). These models capture the intrinsic stochasticity emerging from depth and random initialization, but cannot express the global token mixing of attention or incorporate practical normalization schemes like RMSNorm, which plays a crucial role in preventing explosion and fundamentally alters token dynamics (Karagodin et al., 2024; 2025).

We therefore lack a theoretical framework for Transformers that incorporates *intrinsic* stochasticity, that is noise arising directly from standard random initialization. Understanding this intrinsic noise is essential, as it corresponds to initialization schemes used in real-world applications. In this paper, we analyze a simplified self-attention–only Trans-

former in which randomness enters exclusively through the value matrices at initialization. Under diffusion scaling with RMS normalization, we prove that the discrete token dynamics converge to a system of stochastic differential equations (SDEs) constrained to the unit sphere. Building on this continuous-time limit, we show that intrinsic noise qualitatively alters token clustering: the interplay between attention temperature and token dimension leads to phase transitions that cannot occur in deterministic attention flows. These transitions offer a new perspective on rank collapse phenomena: While noise still drives tokens toward low-rank configurations, it simultaneously diversifies the possible terminal states, enabling pairwise separation.

More precisely, we make the following contributions:

- **Architecturally grounded stochastic limit (Thm. 1).** We prove convergence in distribution of deep randomly initialized Transformers–where noise emerges from random initialization of Value matrices–to a system of SDEs on the sphere. Importantly, the limiting system is invariant to the initialization distribution, as long as it is centered with fixed variance.
- **Explicit phase transition boundaries for $2$ tokens (Thm. 2).** We show that tokens can either converge to a single point or separate into two antipodal points on the sphere—and these are the only possible outcomes. The system can be in one of two regimes, and we derive explicit boundaries for this phase transition in terms of token dimension and the inverse softmax temperature.
- **Noise-induced accessibility of deterministic zero-probability states (Prop. 1).** We introduce noise directly into the deterministic dynamics studied in prior works, establishing a sharp phase transition at an explicit threshold: Below it, deterministic clustering to a single point persists; above it, noise enables tokens to diverge into antipodal configurations.
- **Numerical validation and robustness beyond the analytically tractable regime (Section 6).** We show that random initialization of value matrices leads to an improvement in accuracy on CIFAR-10 compared to constant weight initializations. We present numerical illustrations of predicted phase transitions and clustering behavior. We show that antipodal configurations persist beyond the two-token setting and probe consistency across varying time horizons. Rigorous convergence rates and theoretical guarantees beyond the two-token case remain open directions for future work.

Our analysis concerns signal propagation at initialization and **does not** describe the full training dynamics of Transformers.

**Notations.** Let $\boldsymbol{X} = (X^1, \ldots, X^N)$ collect tokens $X^i \in \mathbb{R}^d$ (column vectors). $P_X^\perp[Y] = Y - \langle X, Y \rangle X$ projects onto the tangent space of $\mathbb{S}^{d-1}$ at $X$. We denote convergence

in distribution by $\xrightarrow{d}$, the space of continuous functions by $C(X; Y)$, and a standard matrix-valued Brownian motion by $W_t \in \mathbb{R}^{d \times d}$. Finally, $\| \cdot \|$ is the Euclidean norm, $\mathcal{N}(0, 1)$ the standard normal, $\mathbb{N}_0$ the non-negative integers, and $\mathrm{Id}(d)$ the identity matrix.

## 2. Background and related work

We briefly review the theoretical frameworks that treat deep networks as continuous-time dynamical systems. These scaling limits provide the mathematical foundation for analyzing signal propagation in deep Transformers.

### 2.1. Residual Neural Networks

Residual networks (ResNets) (He et al., 2016) can be conveniently interpreted as discretizations of differential equations (Chen et al., 2018). Let $X_n \in \mathbb{R}^d$ denote the hidden representation at layer $n$. Consider the layer-wise update

$$X_{n+1} = X_n + \alpha_L V_n f(X_n), \quad n = 0, \ldots, L-1, \quad (1)$$

where $\alpha_L$ is a depth-dependent scaling factor and $V_n \in \mathbb{R}^{d \times d}$ denotes the layer weight matrix, typically initialized randomly before training (e.g., with independent standard normal entries). As the depth $L \to \infty$, the choice of $\alpha_L$ determines the nature of the limiting dynamics, as summarized in Table 1.

*Table 1.* Scaling regimes and their continuous limits. Here, $W_t \in \mathbb{R}^{d \times d}$ is a matrix whose entries are independent scalar Brownian motions.

| $\alpha_L$ | INITIALIZATION | CONTINUOUS LIMIT |
|---|---|---|
| $1/L$ | $V_n = \mathrm{Id}(d)$ | ODE: $dX_t = f(X_t)dt$ |
| $1/\sqrt{L}$ | $V_n^{ij} \overset{\text{I.I.D.}}{\sim} \mathcal{N}(0,1)$ | SDE: $dX_t^\top = f(X_t)^\top dW_t$ |

While the ODE limit ($\alpha_L = 1/L$) describes the flow of infinite-depth networks with near-identity weights (a.k.a. Neural ODE (Chen et al., 2018)), the SDE limit ($\alpha_L = 1/\sqrt{L}$) captures the intrinsic stochasticity arising from random initialization (Yang & Schoenholz, 2017; Cont et al., 2023; Marion et al., 2025). This "diffusion regime" is critical for understanding trainability and gradient stability in deep networks (Peluchetti & Favaro, 2020; Marion et al., 2025). Many works have studied signal propagation in deep residual networks, which include in particular Transformers, e.g. Zhang et al. (2019); Bachlechner et al. (2021); Hayou et al. (2021); Yang et al. (2024); Chizat (2025); Dong et al. (2025). See also references in He & Hofmann (2024). In this work, we study more specifically this stochastic perspective for Transformers, where the analysis is complicated by the global coupling between tokens.

## 2.2. Transformers and Self-Attention

The Transformer architecture (Vaswani et al., 2017) consists of stacked layers combining self-attention mechanisms, feedforward blocks, normalization, and residual connections. The defining component is the attention layer, which updates each *token* (i.e., a vector-valued representation of an element in the input sequence) by aggregating information from the entire sequence.

At an abstract level, the attention mechanism associates to each token $X^i$ a weighted linear combination of the token representations,

$$\mathsf{A}_\beta(X^i, \boldsymbol{X}; \Theta) \ = \ \sum_{j=1}^{N} w_{ij}(X^i, \boldsymbol{X}; \Theta) \, X^j,$$

where the weights $w_{ij}$ depend on pairwise interactions between $X^i$ and $X^j$ and are parameterized by $\Theta$. The precise functional form of these weights is architecture-dependent and will be specified below.

The output of the attention block (see Fig. 7) is then obtained by applying a linear transformation through the Value matrix $V \in \mathbb{R}^{d \times d}$, so that the full update takes the form

$$X^i \ \longmapsto \ V \, \mathsf{A}_\beta(X^i, \boldsymbol{X}; \Theta).$$

**Softmax Self-Attention (SA).** The standard attention mechanism introduced by Vaswani et al. (2017) normalizes exponential interactions via a softmax:

$$\mathsf{A}_\beta^{\mathrm{S}}(X^i, \boldsymbol{X}) := \frac{\sum_{j=1}^{N} \exp\big(\beta \langle QX^i, KX^j \rangle\big) \, X^j}{\sum_{j=1}^{N} \exp\big(\beta \langle QX^i, KX^j \rangle\big)}, \quad (2)$$

where $Q$ and $K$ are respectively the query and key matrices and $\beta$ is an inverse temperature parameter. This formulation constitutes the backbone of modern Transformer architectures and is the primary object of interest from a modeling standpoint.

**Unnormalized Self-Attention (USA).** Alongside SA, we consider its unnormalized counterpart,

$$\mathsf{A}_\beta^{\mathrm{U}}(X^i, \boldsymbol{X}) := \frac{1}{N} \sum_{j=1}^{N} \exp\big(\beta \langle QX^i, KX^j \rangle\big) \, X^j, \quad (3)$$

which has been widely used as a theoretical proxy in the analysis of attention dynamics (Sander et al., 2022; Geshkovski et al., 2023). USA preserves the same exponential interaction geometry as SA, while removing the normalization that couples tokens through the denominator.

Throughout the paper, statements formulated for $\mathsf{A}_\beta \in \{\mathsf{A}_\beta^{\mathrm{S}}, \mathsf{A}_\beta^{\mathrm{U}}\}$ apply to both $\mathsf{A}_\beta^{\mathrm{S}}$ and $\mathsf{A}_\beta^{\mathrm{U}}$.

## 2.3. Related theoretical perspectives

A growing body of work has investigated deep Transformer dynamics through simplified yet mathematically tractable models. Three mainstream theoretical approaches have emerged as particularly relevant to the present work.

**Mean-Field Transformers.** The first approach studies Transformers in a mean-field regime, taking both the network depth and the number of tokens to infinity (Sander et al., 2022; Vuckovic et al., 2021). In this setting, attention dynamics can be expressed in terms of an empirical measure over tokens, leading to deterministic limiting equations that describe smoothness (Castin et al., 2023), universality (Furuya et al., 2025), token sample complexity (Bohbot et al., 2025), clustering (Geshkovski et al., 2023), and metastability (Geshkovski et al., 2024; Bruno et al., 2024; Castin et al., 2025). A key feature of this framework is that attention parameters are typically assumed to be fixed, with the Value matrix treated deterministically. As a result, stochasticity arising from layer-wise random initialization is not captured.

**Stochastic Transformers.** A second line of work introduces stochasticity directly at the level of token dynamics, often by modeling attention-induced interactions through Langevin-type equations (Shalova & Schlichting, 2024; Balasubramanian et al., 2025). In these models, each token is driven by an independent Brownian motion, leading to McKean–Vlasov limits and associated Fokker–Planck equations. While this approach captures important phenomena such as phase transitions, the injected noise is typically *idiosyncratic* and externally imposed, with no direct derivation from the discrete Transformer architecture.

**Signal propagation.**

Another perspective on Transformers comes from the signal propagation literature, which aims at finding parameterizations for which the forward and backward passes are well-conditioned at initialization. Noci et al. (2022) show that the $\alpha_L = 1/\sqrt{L}$ scaling factor in front of the residual connection for attention reduces the magnitude of clustering in the attention layers (which they refer to as *rank collapse*). However, they still empirically observe partial clustering even with the $1/\sqrt{L}$ scaling (see Fig. 4 in their paper). Follow-up works propose Shaped Attention (Noci et al., 2023; He & Hofmann, 2024), which consists in an affine transform of the attention matrix, and obtains competitive results with respect to standard attention. Our work provides theoretical grounding on the rank collapse/clustering phenomenon at initialization with $1/\sqrt{L}$ scaling and i.i.d. Value matrices.

**Position of our work.** In contrast, our paper studies theoretically and numerically stochasticity that arises *intrinsically* from standard random initialization of deep Transformers. Randomness enters exclusively through the Value matrices,

resulting in a diffusion limit driven by *common noise* for all tokens. This modeling choice is close to practical deep Transformers implementations (Touvron et al., 2023; Mistral AI, 2025), where stochasticity arises from layer-wise random initialization rather than from externally injected noise. Therefore, the global coupling induced by attention is preserved, leading to stochastic dynamics that are fundamentally distinct from models with token-wise noise.

## 3. Deep Stochastic Transformers

In this section, we describe the precise Transformer model that underlies our analysis. We work with a minimal yet expressive architecture built from three core components: *(i)* self-attention, *(ii)* residual connections with depth-dependent scaling, and *(iii)* token-wise RMS normalization. Position-wise feedforward blocks are deliberately omitted, as they act independently on each token and therefore do not contribute to inter-token interactions; this allows us to isolate the core mechanism of attention.

### 3.1. Attention Mechanism

As detailed in Section 2.2, we focus on Softmax Self-Attention (SA), defined in (2), and its unnormalized proxy (USA), defined in (3). We recall that throughout the rest of the paper, $\mathsf{A}_\beta$ refers to either formulation unless specified. Importantly, we treat the Key ($K$) and Query ($Q$) matrices as fixed parameters absorbed into the attention function. While standard initialization involves randomness in all weight matrices, freezing $K$ and $Q$ allows the tractable derivation of the continuous-time limit in Section 4. As our results demonstrate, the intrinsic noise arising solely from $V$ is sufficient to break convergence of tokens to a single cluster. We leave the analysis of the fully stochastic setting for future research.

### 3.2. Residual Connections and Scaling

We adopt the *diffusion scaling* $\alpha_L \sim 1/\sqrt{L}$ in the residual update (1), in direct analogy with the scaling regimes introduced in Table 1 for stochastic deep ResNets, leading to nontrivial stochastic limits as $L \to \infty$.

### 3.3. Token-Wise RMS Normalization

This operation constrains each token to the unit sphere, preventing norm explosion and imposing a geometric structure on the dynamics. RMS normalization (*post-layer normalization*) is applied to every token, independently: $\mathrm{RMSNorm}(X^i) = X^i/\|X^i\|$. This normalization projects the attention field onto the tangent space of the sphere by removing the component collinear to each token and preserving only the orthogonal component (see Fig. 1), thereby modifying the induced flow while preserving inter-token

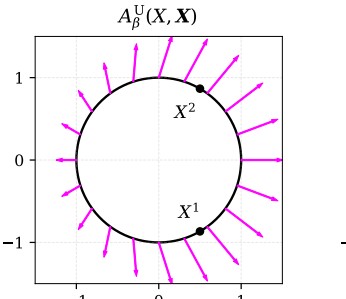
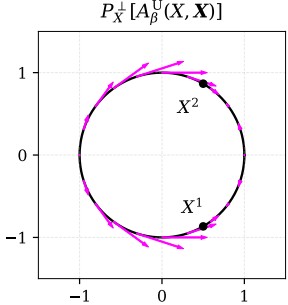

*Figure 1.* Effect of RMS normalization on attention field. Left: unnormalized $\mathsf{A}_\beta^{\mathrm{U}}(X, \boldsymbol{X})$. Right: tangent projection $P_X^\perp[\mathsf{A}_\beta^{\mathrm{U}}(X, \boldsymbol{X})]$ constrains flow to sphere. Here $\boldsymbol{X} = (X, X^1, X^2)$ and $\beta = 1$, $Q = K = \mathrm{Id}(2)$.

interactions. An analysis of normalization-induced geometric effects, with comparisons between different normalization schemes, is provided by Karagodin et al. (2025). RMSNorm is also a standard component of modern LLMs, including LLaMA (Touvron et al., 2023) and Mistral (Mistral AI, 2025).

### 3.4. Deep Stochastic Transformer Model

The complete discrete-time model studied in this work integrates the three architectural elements introduced above: Attention mechanism $\mathsf{A}_\beta$ (Section 3.1), Diffusion-scaled residual connection (Section 3.2), Token-wise RMS normalization (Section 3.3). Arranged as shown in Fig. 7,

$$X_{n+1}^i = \frac{X_n^i + \frac{1}{\sqrt{L}}V_{n+1}\mathsf{A}_\beta(X_n^i, \boldsymbol{X}_n)}{\left\| X_n^i + \frac{1}{\sqrt{L}}V_{n+1}\mathsf{A}_\beta(X_n^i, \boldsymbol{X}_n) \right\|}, n \in \mathbb{N}_0, \quad (4)$$

where $\boldsymbol{X}_n = (X_n^1, \dots, X_n^N) \in (\mathbb{R}^d)^N$, $\mathsf{A}_\beta \in \{\mathsf{A}_\beta^{\mathrm{S}}, \mathsf{A}_\beta^{\mathrm{U}}\}$ and $\{V_n\}_{n \geqslant 1} \subset \mathbb{R}^{d \times d}$ is a sequence of Value matrices, such that:

**Assumption 1.** The matrices $\{V_n\}_{n \geqslant 1} \subset \mathbb{R}^{d \times d}$ are i.i.d. with independent entries $\{v_n^{kl}\}_{k,l=1}^d$ satisfying

$$\mathbb{E}[v_n^{kl}] = 0, \qquad \mathbb{E}\big[(v_n^{kl})^2\big] = \sigma^2, \qquad |v_n^{kl}| \leq M \quad \text{a.s.}$$

for some constant $M > 0$ independent of $n, k, l$.

Note that this assumption is satisfied by standard uniform and truncated normal initializations (Glorot & Bengio, 2010; He et al., 2015).

## 4. Continuous Time Limit

In this section, we describe the continuous-time limit of Deep Stochastic Transformers. The main technical part of our derivation comes from the nonlinearity introduced by RMS normalization.

## 4.1. Continuous-time limit of stochastic transformers

To pass to continuous time, we use the standard linear interpolation for every token trajectory $X^i_\cdot$ and step $n \in \mathbb{N}$:

$$\begin{cases} \Delta X^i_n := X^i_{n+1} - X^i_n, \quad i \in \{1, \ldots, N\}, \\ X^i_L(t) := X^i_n + \frac{t-n/L}{L} \Delta X^i_n, \text{ if } \frac{n}{L} \le t < \frac{n+1}{L}. \end{cases} \quad (5)$$

This yields the continuous-time process

$$\boldsymbol{X}_L(t) = \big( X^1_L(t), \ldots, X^N_L(t) \big) \in C\big( \mathbb{R}_+; (\mathbb{R}^d)^N \big).$$

We can now state the following diffusion limit theorem for deep normalized Transformers.

**Theorem 1** (Continuous-time limit of normalized attention). *Suppose that Assumption 1 holds. Let $\boldsymbol{Y} = (Y^1, \ldots, Y^N)$ be the solution to the **Transformer SDEs**, that is, the system of stochastic differential equations driven by a* common *matrix-valued Brownian motion $W_t \in \mathbb{R}^{d \times d}$:*

$$dY^i_t = \frac{(1-d)\sigma^2}{2} \|\mathsf{A}_\beta(Y^i_t, \boldsymbol{Y}_t)\|^2 Y^i_t \, dt +$$
$$+ \sigma \, P^\perp_{Y^i_t} \big[ dW_t \, \mathsf{A}_\beta(Y^i_t, \boldsymbol{Y}_t) \big], \qquad i = 1, \ldots, N, \quad (6)$$

*with initial condition $Y^i_0 = X^i_0$. Then the linearly interpolated discrete dynamics $\boldsymbol{X}_L$ defined in (5) converge in distribution[1] to $\boldsymbol{Y}$ as $L \to \infty$, namely, $\boldsymbol{X}_L \xrightarrow{d} \boldsymbol{Y}$. The latter convergence holds for $\boldsymbol{Y}$ and $\boldsymbol{X}_L$ viewed as random elements of $C\big( \mathbb{R}_+; (\mathbb{R}^d)^N \big)$.*

The proof of this result relies on the limiting properties of the nonlinear difference equation (4) and their characterization via a martingale problem (Kushner & Huang, 1981; Watanabe, 1984a;b; 1988). We defer the proof to App. B.3.

Thm. 1 is reminiscent of the celebrated Donsker invariance principle (Donsker, 1951), which states that random walks, when properly scaled, converge weakly to Brownian motions. In this context, we treat our discrete scheme (4) as a modified random walk on $(\mathbb{S}^{d-1})^N$. It is important to note that, unlike strong convergence, we do not require Gaussian steps. In fact, regardless of the specific random initialization, all will converge to the same limit as $L \to \infty$, provided the second moment is fixed, entries are centered and the support is bounded. This ensures convergence for **any** centered initialization of the Value matrices, with no need for Gaussian initialization.

It is worth saying a few words about the additional drift term in equation (6). A naive approach to stochasticizing the deep Transformer would be to introduce a matrix-valued Brownian motion directly into the continuous-time attention dynamics on the sphere:

$$dX^i_t = P^\perp_{X^i_t}[dW_t \mathsf{A}_\beta(X^i_t, \boldsymbol{X}_t)],$$

_______________
[1]Convergence in distribution is defined in App. B.2, see (12).

in direct analogy with ResNets (see Table 1). However, such an equation does not preserve the spherical constraint. Even if the noise is tangential, applying Itô's formula to the evolution of $\|X^i_t\|^2$ given by the formula above reveals a nonzero radial drift generated by second-order terms, causing trajectories to leave the sphere. This is a classical phenomenon in the theory of spherical Brownian motion (Stroock, 1971). As a simple illustration, consider $X_t = (X^1_t, X^2_t) \in \mathbb{S}^1$ evolving according to $dX_t = \frac{1}{\|X_t\|}(-X^2_t, X^1_t) \, dW_t$. Although $\langle X_t, dX_t \rangle = 0$, Itô's formula yields $d\|X_t\|^2 = dt$, so $X_t$ immediately leaves the sphere. By contrast, we have the following lemma.

**Lemma 1.** *For any sphere-supported initial distribution $\boldsymbol{Y}_0 \in (\mathbb{S}^{d-1})^N$, any solution of equation (6) remains on the unit sphere with probability 1.*

## 4.2. Bridging Deterministic and Stochastic Dynamics

Beyond initialization, training naturally leads to dynamics that combine a deterministic component with a stochastic one. The intuition that training adds a deterministic component with strong correlations across successive layers is supported by numerical experiments in Cohen et al. (2021); Marion et al. (2025) for i.i.d. initialization, and by both numerical experiments and proofs in Marion et al. (2023) in the weight-tied setting. The hybrid model considered in this section, where the deterministic component corresponds to an identity value matrix shared across all layers, should be viewed as a first step toward understanding signal propagation after initialization.

To precisely trace how intrinsic noise alters the clustering behavior, we study a parametrized perturbation of the deterministic dynamics. We therefore introduce a *hybrid model*, in which the update step combines a deterministic drift of order $1/L$ with a stochastic perturbation of order $\varepsilon/\sqrt{L}$:

$$X^i_{n+1} = \frac{X^i_n + \omega_\varepsilon(n, L) \mathsf{A}_\beta(X^i_n, \boldsymbol{X}_n)}{\|X^i_n + \omega_\varepsilon(n, L) \mathsf{A}_\beta(X^i_n, \boldsymbol{X}_n)\|}. \quad (7)$$

Here $n \in \mathbb{N}_0$, $X^i_0 \in \mathbb{S}^{d-1}$, and

$$\omega_\varepsilon(n, L) = \tfrac{1}{L} \bar{V} + \varepsilon \tfrac{v_n}{\sqrt{L}} \operatorname{Id}(d),$$

where $\{v_n\} \subset \mathbb{R}$ are i.i.d. centered random variables with unit variance and bounded support, the matrix $\bar{V} \in \mathbb{R}^{d \times d}$ is constant, and $\varepsilon \ge 0$ controls the noise amplitude; $\varepsilon = 0$ corresponds to the deterministic case. Its continuous-time limit makes explicit how noise enters the drift and qualitatively alters the long-time clustering behavior.

The following two lemmas characterize this limit. The first shows that for $\varepsilon = 0$, we recover the deterministic attention flow studied in Geshkovski et al. (2025).

**Lemma 2.** *When $\varepsilon = 0$, the rescaled dynamics (7) converge*

*uniformly to the solution of*

$$dY_t^i = P_{Y_t^i}^\perp \left[ \bar{V} \mathsf{A}_\beta(Y_t^i, \boldsymbol{Y}_t) \right] dt, \qquad Y_0^i = X_0^i.$$

*That is,* $\sup_{i,n} \| X_n^i - Y_{n/L}^i \| \to 0$ *as* $L \to \infty$, *where the supremum is taken over* $1 \le i \le N$ *and* $0 \le n \le L - 1$.

The second lemma gives the stochastic limit for $\varepsilon > 0$, analogous to Thm. 1.

**Lemma 3.** *Let* $\boldsymbol{X}_L^\varepsilon(t)$ *be the linear interpolation of* (7) *and let* $\boldsymbol{Y}_t^\varepsilon$ *solve the SDE*

$$dY_t^i = \left[ \varepsilon^2 \mathcal{F}\left(Y_t^i, \mathsf{A}_\beta(Y_t^i, \boldsymbol{Y}_t)\right) + P_{Y_t^i}^\perp \left[\bar{V} \mathsf{A}_\beta(Y_t^i, \boldsymbol{Y}_t)\right] \right] dt \\ + \varepsilon\, P_{Y_t^i}^\perp \left[\mathsf{A}_\beta(Y_t^i, \boldsymbol{Y}_t)\right] dB_t, \quad (8)$$

*where* $\mathcal{F}(x,y) := \frac{3}{2}\langle x, y \rangle^2 x - \frac{1}{2}\|y\|^2 x - \langle x, y \rangle\, y$, *and* $B_t$ *is a scalar Brownian motion. Then*

$$\boldsymbol{X}_L^\varepsilon \xrightarrow{d} \boldsymbol{Y}^\varepsilon \quad \text{as } L \to \infty.$$

*This convergence holds for* $\boldsymbol{Y}^\varepsilon$ *and* $\boldsymbol{X}_L^\varepsilon$ *viewed as random elements of* $C(\mathbb{R}_+; (\mathbb{R}^d)^N)$.

This limiting SDE explicitly shows how the noise amplitude $\varepsilon$ reweights the drift (through $\varepsilon^2 \mathcal{F}$) and adds a diffusion term. In Section 5.3, we analyze the long-time behavior of (8) and demonstrate that a sufficiently large $\varepsilon$ provably switches the clustering outcome from a single point to the antipodal configuration.

# 5. Clustering

In this section, we begin by defining the notion of clustering in the stochastic setting. We then state the main results concerning the long-time behavior of Transformer SDEs and highlight the relationship between $(\beta, d)$ and the clustering behavior in the case $N = 2$.

## 5.1. Notion of Clustering

We first define clustering configurations for two tokens.

**Definition 1.** *Let* $Y^i$ *and* $Y^j$ ($i \ne j$) *be two tokens among* $N$ *governed by the dynamics* (6). *We define* **clustering to a single token** *as the following random event:*

$$A_1^{ij}: \quad \left\| Y_t^i - Y_t^j \right\| \to 0 \text{ as } t \to \infty. \qquad (9)$$

Similarly, we define **clustering to two antipodal tokens** as the following random event:

$$A_2^{ij}: \quad \left\| Y_t^i + Y_t^j \right\| \to 0 \text{ as } t \to \infty. \qquad (10)$$

When $N = 2$, we denote $A_1 = A_1^{12}$ and $A_2 = A_2^{12}$.

## 5.2. On the Relation Between $\beta$ and $d$ with Clustering

When the number of tokens is restricted to $N = 2$, the long-time behavior of the Transformer SDEs can be analyzed explicitly. To obtain closed-form phase-transition thresholds in terms of $(\beta, d)$, we introduce the following simplifying assumption, which is used *only* for this explicit calculation.

**Assumption 2.** For the purpose of explicit phase-boundary computations, the key and query matrices satisfy $Q^\top K = \mathrm{Id}(d)$.

This identity specialization simplifies the interaction geometry and allows for closed-form expressions. Importantly, it is *not required* for the existence of the continuous-time limit or for the general formulation of the Transformer SDEs.

Under this assumption, explicit phase transitions can be characterized in the two-token case.

**Theorem 2.** *Let* $N = 2$, *let the tokens follow* (6), *and suppose that* $\langle Y_0^1, Y_0^2 \rangle \notin \{-1, +1\}$ *a.s. Suppose that Assumption 2 holds. Then,* $\mathbb{P}(A_1) + \mathbb{P}(A_2) = 1$. *Moreover,* $\mathbb{P}(A_1) > 0$ *and furthermore:*

- $d - 2 \ge \cosh(2\beta)$ *implies* $\mathbb{P}(A_2) = 0$,
- $d - 2 < \cosh(2\beta)$ *implies* $\mathbb{P}(A_2) > 0$.

The argument proceeds as follows. We first derive an SDE for the overlap diffusion by exploiting the stochastic limit obtained in Thm. 1. This diffusion is a one-dimensional Markov process adapted to the same filtration, and moreover its diffusion coefficient is strictly positive on the interval $(-1, 1)$. Thus, the theorem reduces to the Feller classification of boundary points for one-dimensional diffusions. The detailed proof is deferred to App. C.3.

**Contrast with the deterministic case.** This phase transition is a direct consequence of intrinsic noise. In the deterministic setting with $K = Q = \mathrm{Id}(d)$, tokens always converge to a single cluster for any $\beta, d$ (Criscitiello et al. (2024), see also Thm 6.1 in (Geshkovski et al., 2025)). Here, stochasticity unlocks a new regime: for $\beta > \frac{1}{2}\cosh^{-1}(d - 2)$, the antipodal configuration emerges as a possibility.

Importantly, Thm. 2 provides a *partial* solution to Problem 2.15[2] posed by Geshkovski et al. (2025), asking whether the deterministic clustering conclusions could be generalized to Transformers with random matrices $(Q, K, V)$.

## 5.3. Noise-Induced Clustering

The continuous-time limit of the hybrid model, given by Lem. 3, provides a transparent setting to quantify how noise reshapes clustering. For the two-token case, we can derive an explicit threshold that separates the single-cluster and antipodal regimes.

---

[2]Corresponds to Problem 3 in the arXiv preprint version.

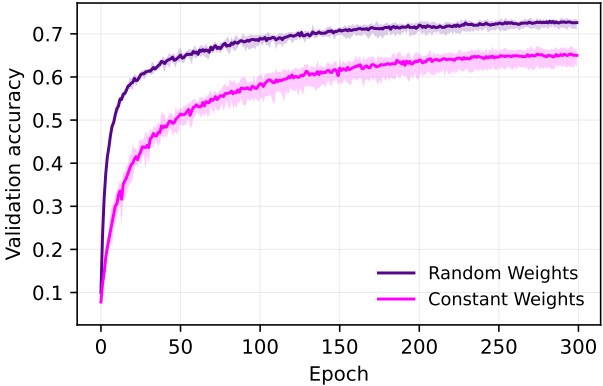

*Figure 2.* Effect of value-matrix initialization on CIFAR-10 validation accuracy. **Constant Weights** (magenta) correspond to *identity initialization* of the matrices $V$ shared across all layers, while **Random Weights** (violet) correspond to *independent Gaussian initialization* of $V$ in each layer. Curves show the mean validation accuracy over five independent training runs; shaded regions indicate the min. and max. accuracy across runs at each epoch.

**Proposition 1** (Noise-induced clustering transition). *Let* $\mathsf{A}_\beta = \mathsf{A}_\beta^{\mathrm{U}}$, $N = 2$, $\bar{V} = \mathrm{Id}(d)$ *and Assumption 2 hold, and suppose that* $\langle Y_0^1, Y_0^2 \rangle \notin \{-1, +1\}$ *a.s. The solution of the limiting SDE (8) satisfies:*
**1.** *If* $\varepsilon^2 < 2e^{-\beta}$, $\lim_{t\to\infty} \|Y_t^1 - Y_t^2\| = 0$, *with probability 1, i.e., the tokens converge to a single cluster.*
**2.** *If* $\varepsilon^2 > 2e^{-\beta}$, $\lim_{t\to\infty} \|Y_t^1 + Y_t^2\| = 0$, *with probability 1, i.e., the tokens converge to the antipodal configuration.*

Prop. 1 delivers a sharp, interpretable condition: a sufficiently large noise amplitude $\varepsilon$ (relative to the inverse temperature $\beta$) forces the system into the antipodal phase. This result complements Thm. 2 obtained for the full stochastic transformer. Numerical simulations (Fig. 5) confirm the theoretical threshold $\varepsilon_c(\beta) = \sqrt{2e^{-\beta}}$ and illustrate how the noise-controlled transition manifests in the discrete dynamics (7). Together, these findings underscore that the stochasticity inherent in random initialization is not a technical detail, but qualitatively alters the clustering landscape of deep transformers.

As shown in App. C.1, the overlap diffusion—namely, the one-dimensional diffusion governing the pairwise inner product of two tokens—reveals why antipodal configurations are reachable. In addition, apart from coincidence and antipodality, the diffusion is strictly positive, making intermediate states unstable. Near the boundaries, both drift and diffusion vanish linearly.

## 6. Experimental Results

**Key Role of Randomness.** To legitimize our focus on random initialization, we empirically demonstrate that such initialization in deep self-attention stacks leads to better ac-

curacy compared to weight-tied initialization, using a simplified Vision Transformer trained on CIFAR-10 (Krizhevsky et al., 2009). Our implementation is based on a public PyTorch reimplementation of Vision Transformer for CIFAR datasets,[3] following the original ViT architecture (Dosovitskiy et al., 2021) and using AutoAugment data augmentation (Cubuk & Zoph, 2018). The model consists of 128 self-attention layers with embedding dimension $d = 96$ and RMS normalization. All models are trained for 300 epochs with batch size 128 and linear learning-rate warmup over 8 epochs, and a cosine learning-rate decay schedule thereafter. Results are averaged over 5 random seeds. The key and query matrices $K$, $Q$ are initialized once, shared identically across all layers, and remain fully trainable throughout optimization. We consider a single-head attention architecture without any feed-forward (MLP) layers, so that each block consists solely of attention followed by normalization, exactly as in (4). The model is trained end-to-end using the standard cross-entropy loss.

We compare two initializations for the value matrices $V$: **(i) constant identity** initialization, where all layers share the same value matrix initialized as the identity, and **(ii) independent Gaussian** initialization, where each layer's value matrix is initialized independently with i.i.d. Gaussian entries of standard deviation $\sigma = 1$. Other architectural choices, hyperparameters, and training settings are fixed. Despite identical architectures and fully trainable parameters, identity-initialized $V$ lead to a degradation in validation accuracy. This indicates that independence at initialization has a significant impact on the training dynamics of deep attention stacks. Note that a similar observation is made by He & Hofmann (2024, Fig. 23) in a language task.

**Numerical Verification of the Phase Transition.** We numerically investigate the phase transition predicted by Thm. 2 by directly simulating the *discrete Transformer dynamics* (4). Importantly, no time-discretization of the limiting SDE (6) is used; all experiments are performed with the original discrete-time model, rather than discretizing the continuous-time stochastic differential equation using methods like Euler-Maruyama. This distinction is crucial, as the properties we establish for the continuous-time dynamics, particularly with respect to behavior at infinity, are not guaranteed to hold in the discrete case.

We consider $N = 2$ tokens and dimensions $d \in \{4, \ldots, 10\}$. For each pair $(\beta, d)$, we simulate $40{,}000$ independent trajectories, initialized uniformly on the unit sphere. The discrete dynamics (4) are simulated with $L = 100$ and $T = 500$. At each layer, the value matrix $V_n$ is independently re-initialized with i.i.d. Gaussian entries of variance $\sigma^2 = 1$, while the key and query matrices are fixed and equal to the identity, $K = Q = \mathrm{Id}(d)$. Clustering is de-

---
[3]https://github.com/omihub777/ViT-CIFAR

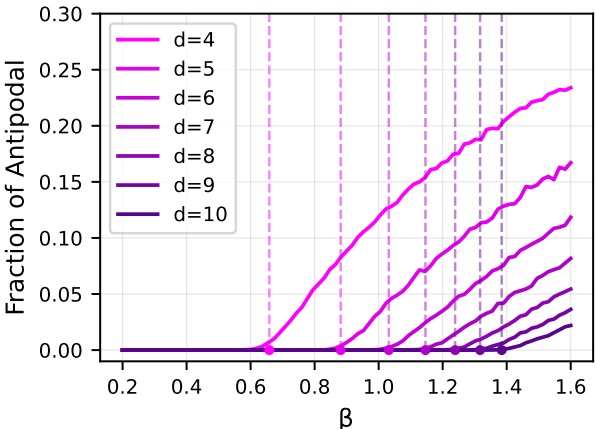

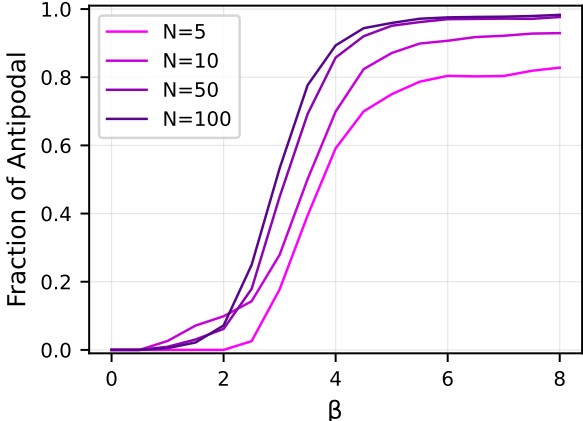

*Figure 3.* **Noise-induced phase transition for two tokens.** Empirical probability of antipodal convergence as a function of $\beta$ for dimensions $d \in \{4, \ldots, 10\}$. Each point is estimated from 40,000 independent trajectories of the discrete dynamics (4) with $L = 100$, the time horizon $T = 500$, and $\sigma = 1$. The dashed curve shows the theoretical phase boundary $\beta_c(d) = \frac{1}{2}\cosh^{-1}(d-2)$.

*Figure 4.* **Empirical probability of observing at least one antipodal pair in the terminal configuration** as a function of $\beta$, for multiple values of $N$ at dimension $d = 4$. Simulations are performed on a uniform grid $\beta \in [0, 8]$ with step size 0.5, using 10,000 independent trajectories per parameter setting and a fixed time horizon $T = 500$.

tected at the final layer using inner-product thresholds: a trajectory is classified as *antipodal* if $\langle X_T^1, X_T^2 \rangle \leq -1 + \varepsilon$, with $\varepsilon = 10^{-3}$. Empirical probabilities are obtained by averaging over all trajectories.

Fig. 3 shows the empirical probability of antipodal convergence as a function of $\beta$ for each dimension $d$. A sharp transition is observed near the theoretical boundary $\beta_c(d) = \frac{1}{2}\cosh^{-1}(d-2)$ predicted by the continuous-time analysis. This demonstrates that the phase transition derived for the limiting SDE accurately describes the long-time behavior of the *discrete* Transformer dynamics. We checked that the results are not sensitive to varying the time horizon $T$ (see App. D.1 for the details).

**Multiple Tokens Case.** We extend the numerical study of the antipodal clustering regime to systems with $N \geq 3$ tokens. All experiments use a fixed depth $L = 100$ and time horizon $T = 500$, and are based on 10,000 independent trajectories per parameter setting. For $d = 4$, we run simulations on a uniform grid $\beta \in [0, 8]$ with step size 0.5 and varying $N$, and record the empirical probability that the terminal configuration contains at least one antipodal pair. As shown in Fig. 4, this probability becomes strictly positive across different values of $N$, indicating that antipodal structure is not specific to the two-token case. Beyond mere occurrence, we emphasize *persistence*. For large $\beta$, we observe that clustering typically occurs well before $T = 500$, after which the fraction of trajectories exhibiting antipodal structure remains stable over long times (Fig. 8), indicating long-lived antipodal states of the noisy dynamics. The stability of the antipodal configuration for more than two tokens remains an open question, though we conjecture that it is indeed stable, rather than merely metastable. We additionally conjecture that, beyond the single-cluster and antipodal regimes, no other stable asymptotic configurations occur. Numerical experiments (see Fig. 9) are consistent with this picture, although a detailed theoretical and numerical investigation is left for future work.

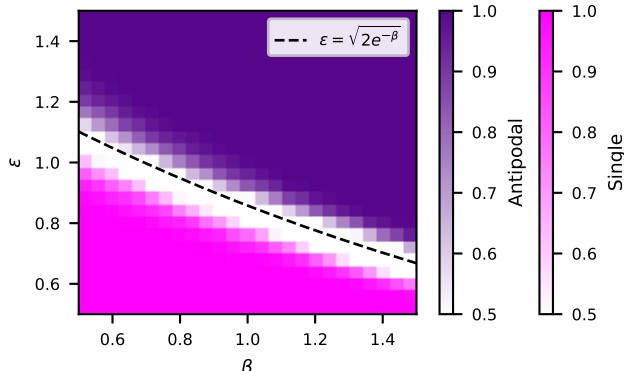

*Figure 5.* **Noise-induced clustering threshold.** Dominant clustering outcome of the discrete hybrid dynamics in the $(\beta, \varepsilon)$ plane. Colors encode the maximum of the empirical fractions of single-cluster (magenta) and antipodal (violet) configurations. Dashed curve $\varepsilon = \sqrt{2e^{-\beta}}$: theoretical threshold from Prop. 1.

**Numerical verification of noise-induced threshold.** We illustrate the noise-induced clustering transition predicted by Prop. 1 using simulations of the discrete hybrid dynamics (7). Fig. 5 shows a phase diagram in the $(\beta, \varepsilon)$ plane obtained from $10^3$ independent trajectories with dimension $d = 3$, depth $L = 100$, and final time $T = 50$. For each pair $(\beta, \varepsilon)$, we compute the empirical fractions of trajectories converging to a single cluster and to an antipodal config-

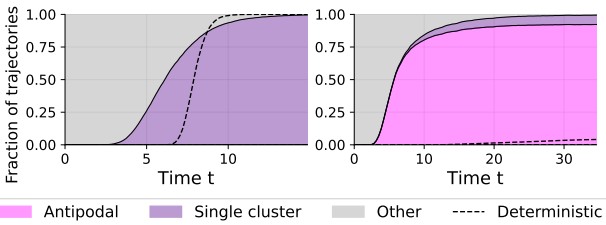

*Figure 6.* Comparison of convergence behavior in two representative regimes. **Left:** $\beta = 0.5$, number of tokens $N = 50$, time horizon $T = 15$, with standard discretization ($L = 100$). **Right:** $N = 50$, $\beta = 4.5$, time horizon $T = 35$, with the same discretization. In both cases, an antipodal configuration is detected when all pairwise inner products satisfy $|\langle x_i, x_j \rangle| \geq 1 - \varepsilon$ and at least one pair satisfies $\langle x_i, x_j \rangle \leq -1 + \varepsilon$, with $\varepsilon = 10^{-3}$.

uration. The heatmap displays the *maximum* of these two fractions, with color indicating the dominant outcome (magenta: single cluster, violet: antipodal). The dashed curve $\varepsilon = \sqrt{2e^{-\beta}}$ is the theoretical threshold derived in Prop. 1. A sharp transition is observed across this curve, confirming that the analytically predicted threshold accurately captures the clustering behavior of the discrete dynamics.

**Rate of convergence and metastable behavior.** While a detailed analysis of metastability and convergence rates is beyond the scope of the present work, we briefly comment on the qualitative behavior observed numerically, focusing on two representative regimes of the inverse temperature parameter $\beta$. All results are averaged over 10,000 independent trajectories; see Figure 6.

- **Small-$\beta$ regime (comparable convergence rates).** In this regime, both dynamics converge predominantly to a single-cluster configuration, with comparable observed convergence rates. The deterministic dynamics is the specialization of (7) with $\varepsilon = 0$, $\mathsf{A}_\beta = \mathsf{A}_\beta^{\mathsf{S}}$, $Q^\top K = \mathrm{Id}(4)$.
- **Large-$\beta$ regime (absence of metastability in the stochastic dynamics).** Both dynamics still converge to a clustered configuration, but their behavior differs markedly: the deterministic dynamics exhibits metastable behavior with long-lived transient configurations, whereas no such metastability is observed in the stochastic dynamics, which converges significantly faster.

Overall, the stochastic dynamics does not display metastable behavior in the sense of Geshkovski et al. (2024), in contrast to its deterministic counterpart at large values of $\beta$.

## 7. Discussion

**Practical relevance.** Despite the theoretical nature of this work, our analysis suggests several insights relevant to practical Transformer architectures:

- Contrary to popular belief that stochasticity helps to reg-

ularize the dynamics, our theory and experiments show that the effective rank of the tokens still decreases in deep stochastic self-attention networks. In particular, initialization noise in the value matrices is insufficient to prevent collapse.
- As the number of layers increases, the limiting dynamics for constant $K, Q$ and random $V$ depend on the initialization of $V$ only through its first two moments. This suggests that, in sufficiently deep Transformers, the precise choice of initialization distribution for the value matrices may become unimportant, provided the first two moments are preserved.

**Limitations and technical obstacles.** We briefly summarize the main technical difficulties; further details can be found in the remarks in the corresponding appendices.

1. **Clustering for $N \geqslant 3$ tokens.** The pairwise overlaps no longer form a closed one-dimensional diffusion. Instead, one must analyze a coupled system of $N(N-1)/2$ strongly correlated stochastic equations, where the expected limiting behavior involves concentration near several degenerate configurations. The analysis is further complicated compared to the deterministic case because the process dynamics no longer strictly minimize the energy functional $\mathsf{E}_\beta$ proposed in (Geshkovski et al., 2025), and the limiting equation for the density of the token system as $N \to \infty$ becomes a PDE with noise.
2. **Random $K, Q$.** Unlike $V$, the behavior in the case of random $K$ and $Q$ depends essentially on the specific initialization of the matrices $K, Q$. Heuristically, in the case when the matrices $Q, K$ and $V$ are independent the limit is determined by the first two moments of $V$ and the exponential moment of $Q^\top K$. Consequently, an analysis analogous to ours is feasible for random $K, Q$, but must be tailored to each specific case. In this regard, a detailed classification of the distributions of $Q, K$ from the perspective of clustering and its rate remains an open theoretical problem.

**Future directions.** All of the technical obstacles above may be interpreted as directions for future work. We also emphasize the importance of carefully studying training dynamics and their impact on the initial stochasticity.

**Conclusion.** In this paper, we studied deep Transformer dynamics in a regime where stochasticity arises *intrinsically* from standard random initialization of value matrices. By deriving a diffusion limit under RMS normalization, we showed that initialization-induced noise qualitatively changes clustering behavior, unlocking antipodal configuration regimes that are provably inaccessible in deterministic attention flows and leading to sharp phase transitions governed by the attention temperature and dimension.

## Acknowledgments

The authors are grateful to Gabriel Peyré for fruitful discussions. We thank Valentin De Bortoli for comments on a draft of this paper. We also thank the anonymous reviewers for their valuable feedback and suggestions. This work was supported in part through the NYU Shanghai IT High Performance Computing resources, services, and staff expertise. L. F. acknowledges support from the MacCracken Fellowship, under which this research was conducted. P.M. is supported by a Google gift.

## Impact statement

This paper analyzes theoretical properties of deep Transformers where stochasticity arises from random initialization of value matrices. We do not foresee any specific ethical or societal implications arising directly from this work.

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

# A. Additional Figures

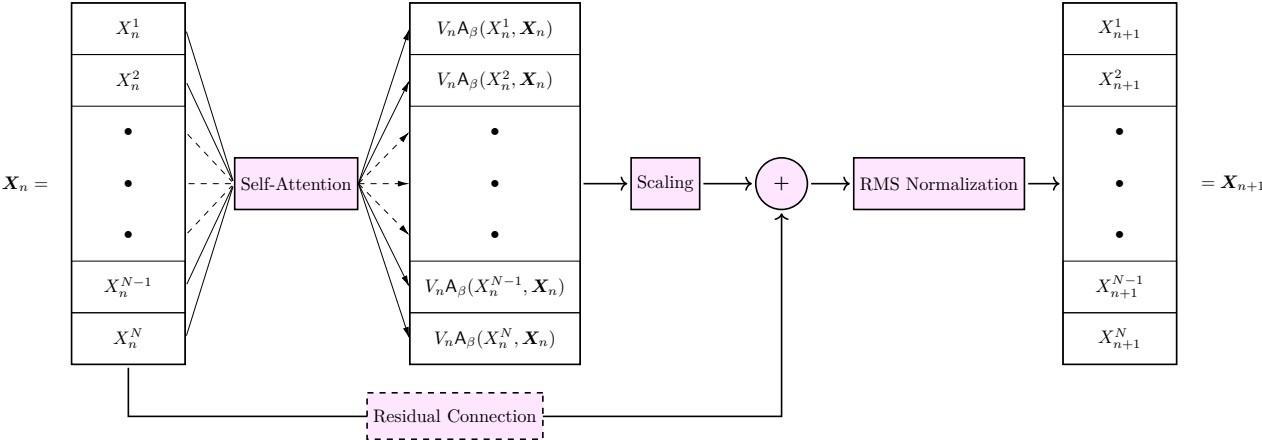

*Figure 7.* One layer of the Deep Transformer with added normalization and diffusion scaling ($\sim 1/\sqrt{L}$). The diagram highlights that self-attention acts globally across all tokens. The residual connection is shown with a dashed outline to indicate that it represents a mode of connectivity rather than an operation.

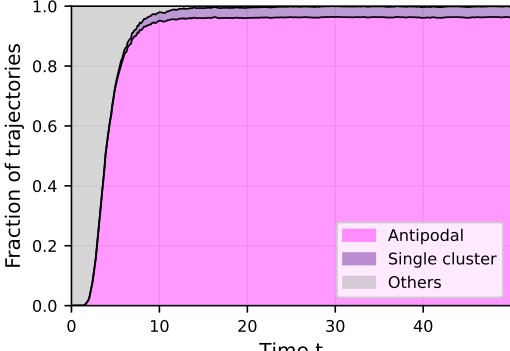

*Figure 8.* Time evolution of clustering fractions for a multi-token system ($d = 4$, $N = 50$) at inverse temperature $\beta = 5$. All curves are obtained by averaging over 2,000 independent trajectories and are shown up to time horizon $T = 50$.

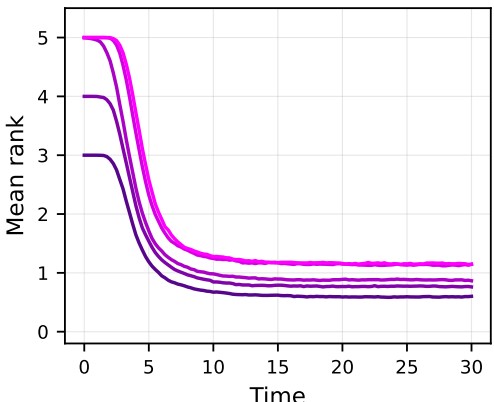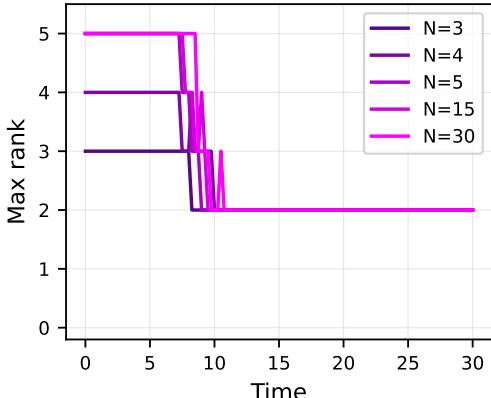

*Figure 9.* Time evolution of the rank of the token system for different numbers of tokens, with $d = 5$ and inverse temperature $\beta = 5$. The left panel shows the mean rank across trajectories, while the right panel shows the maximum rank observed across trajectories. We provide supplementary evidence to Figures 5 and 6 of the main text: despite the persistence of antipodal configurations for large $\beta$, the average and maximum ranks of the token matrix do not scale with $N$, confirming that rank collapse persists in the presence of injected noise. All curves are obtained by averaging over 2000 independent realizations under the assumptions $Q^\top K = I_d$ and i.i.d. Gaussian value matrices $V_{ij} \sim \mathcal{N}(0, 1)$. We use a numerical rank threshold of $10^{-3}$, where a rank of 0 denotes a collapsed state in which all tokens have converged to a single point.

## B. Continuous Time Limit

First, we fix the basic notion of weak convergence used throughout this appendix (Appendix B.2). We then establish the continuous-time limit of the discrete Transformer dynamics under diffusion scaling, proving weak convergence to the limiting stochastic differential equation stated in Theorem 1 (Appendix B.3). Finally, we show that the limiting dynamics preserve the spherical constraint almost surely, so that trajectories remain on the unit sphere for all times (Lemma 1). At the end, we provide the proofs of Lemmas 2 and 3.

### B.1. Notation Convention

The space of infinitely smooth functions from $X$ to $Y$ is denoted by $C^\infty(X, Y)$. Here we deal with $(\mathbb{R}^d)^N$–valued diffusion processes and to simplify the notation, we treat $\boldsymbol{X} \in (\mathbb{R}^d)^N$ as column vector $\boldsymbol{X} = \begin{bmatrix} X^1 \\ \vdots \\ X^N \end{bmatrix}$, where $X^i \in \mathbb{R}^d$ is a column vector as well. Vector calculus notation for the gradient $\nabla\phi(\boldsymbol{X})$ and Hessian $\nabla^2\phi(\boldsymbol{X})$ changes accordingly.

### B.2. Background on Martingale Problem and Weak Convergence

We do not aim at a complete or self-contained exposition of the probabilistic framework used in this work. Instead, we follow the general setting developed by Stroock and Varadhan and only recall the basic definitions and notation that will be used throughout the paper. For a detailed and systematic treatment, we refer the reader to the classical references on weak convergence and stochastic processes (Stroock & Varadhan, 2007; Billingsley, 2013; Bogachev, 2018).

Consider the *canonical probability space* $\Omega = C(\mathbb{R}_+; \mathbb{R}^D)$ of continuous functions taking values in $\mathbb{R}^D$, equipped with the metric

$$d(\omega, \omega') := \sum_{n=0}^{+\infty} \frac{1}{2^n} \frac{\sup_{0 \le t \le n} \|x(t, \omega) - x(t, \omega')\|}{1 + \sup_{0 \le t \le n} \|x(t, \omega) - x(t, \omega')\|}, \tag{11}$$

where $x(t, \omega) := \omega(t)$ denotes the position of the trajectory $\omega$ at time $t$. With this metric, $(\Omega, d)$ is a Polish[4] space. Endowed with its Borel $\sigma$-algebra $\mathcal{B}(\Omega)$, this is the underlying measurable space on which we work.

Let $\{\mu_n\}_{n=1}^\infty \subset \mathcal{P}(\Omega)$ be a sequence of probability measures. We say that $\mu_n$ *converges weakly* to $\mu \in \mathcal{P}(\Omega)$, and write

$$\mu_n \xrightarrow{w} \mu,$$

---

[4]That is, complete and separable; see Bogachev (2018).

if for every bounded continuous function $f \colon \Omega \to \mathbb{R}$,

$$\int f(\omega) \, \mu_n(d\omega) \;\longrightarrow\; \int f(\omega) \, \mu(d\omega), \qquad \text{as } n \to \infty.$$

If $X_n$ and $X$ are $\Omega$-valued random variables such that $\mu_n = \mathrm{Law}(X_n)$, $\mu = \mathrm{Law}(X)$, then the above convergence is called *convergence in distribution*. In this case, we write

$$X_n \xrightarrow{d} X. \tag{12}$$

**Martingale problem.**  We recall the notion of the martingale problem in the sense of Stroock and Varadhan, adapted to the present setting. We emphasize that, although one may formulate the martingale problem together with an explicit filtration, in this work we do *not* specify a filtration a priori. Instead, throughout the paper we implicitly work with the natural filtration generated by the coordinate process.

Let $E = \mathbb{R}^D$ and let $\Omega$ be the canonical path space. Let $L$ be a (second-order) diffusion operator on $E$ with domain $\mathrm{dom}(L) \subset C_b(E)$, and let $\nu \in \mathcal{P}(E)$ be a prescribed initial distribution.

**Definition 2.**  A probability measure $P \in \mathcal{P}(\Omega)$ is said to be a *solution of the martingale problem for* $(L, \nu)$ if

1. the initial marginal of $P$ is $\nu$, that is
$$P \circ X_0^{-1} = \nu;$$

2. for every $f \in \mathrm{dom}(L)$, the process
$$M_f(t) := f(X_t) - f(X_0) - \int_0^t Lf(X_s) \, ds, \qquad t \geq 0,$$
   is a $P$-martingale with respect to the natural filtration.

In this case, we write $P \in \mathrm{MP}(L, \nu)$.

### B.3. Continuous-Time Limit of Deep Stochastic Transformers

*Proof of Theorem 1.*  First, we carry out some preliminary steps. Note that, to establish weak convergence on $\Omega$, it suffices to prove it on an arbitrary but fixed finite time horizon $T \in \mathbb{R}_+$. This follows directly from the form of the metric (11). We therefore fix a time horizon $T$. We decompose the proof into the following steps:

1. Linearize the dynamics (4) and show that the linearized dynamics converges to the nonlinear one;

2. Show that the limit of the linearized dynamics is given by (6);

3. Transfer the convergence result from the linearized dynamics to the original nonlinear dynamics.

Now we implement the steps outlined above. For this purpose, we introduce an auxiliary function $\tilde{\mathsf{A}}_\beta \colon \mathbb{R}^d \times \left(\mathbb{R}^d\right)^N \to \mathbb{R}^d$ such that $\tilde{\mathsf{A}}_\beta \in C^\infty\left(\mathbb{R}^d \times \left(\mathbb{R}^d\right)^N, \mathbb{R}^d\right)$ and

$$\tilde{\mathsf{A}}_\beta(X, \boldsymbol{X}) := \begin{cases} \mathsf{A}_\beta, & \text{if } \|X\|, \|X^i\| \leq 2 \text{ for } 1 \leq i \leq N, \\ 0, & \text{if } \|X\|, \|X^i\| \geq 4. \end{cases}$$

Clearly, this function is not unique, coincides with $\mathsf{A}_\beta$ on $\mathbb{S}^{d-1} \times \left(\mathbb{S}^{d-1}\right)^N$, and is globally bounded. We note that any result proved for the recurrent relation below

$$X_{n+1}^i = \frac{X_n^i + \frac{1}{\sqrt{L}} V_{n+1} \, \tilde{\mathsf{A}}_\beta(X_n^i, \boldsymbol{X}_n)}{\left\| X_n^i + \frac{1}{\sqrt{L}} V_{n+1} \, \tilde{\mathsf{A}}_\beta(X_n^i, \boldsymbol{X}_n) \right\|}, \, n \in \mathbb{N}_0, \boldsymbol{X}_0 \in \left(\mathbb{S}^{d-1}\right)^N, \tag{13}$$

also holds for the recurrent relation (4), since the corresponding solutions coincide. At this stage, the introduction of the modified function merely allows us to state results on the entire space without additional comments, while they automatically apply to the compact subset of interest.

**Step 1. [Linearization]** We note that the following expansion is valid for $X \in \mathbb{S}^{d-1}$ and $\boldsymbol{X} \in \left(\mathbb{S}^{d-1}\right)^N$:

$$\frac{X + \delta V \tilde{\mathsf{A}}_\beta(X, \boldsymbol{X})}{\left\| X + \delta V \tilde{\mathsf{A}}_\beta(X, \boldsymbol{X}) \right\|} = X + \delta P_X^\perp [V \tilde{\mathsf{A}}_\beta(X, \boldsymbol{X})] + \delta^2 G(X, V \tilde{\mathsf{A}}_\beta(X, \boldsymbol{X})) + \delta^3 R(X, V \tilde{\mathsf{A}}_\beta(X, \boldsymbol{X})),$$

where $G$ is given by

$$G(x, y) := \tfrac{3}{2}\langle x, y \rangle^2 x - \tfrac{1}{2}\langle y, y \rangle x - \langle x, y \rangle y,$$

and $R(X, V \tilde{\mathsf{A}}_\beta(X, \boldsymbol{X}))$ is a uniformly bounded remainder term. Discarding the remainder, we obtain a *truncated* discrete-time dynamics $\{Y_n^i\}_{n \in \mathbb{N}_0} \subset \mathbb{R}^d$ with $\delta = \frac{1}{\sqrt{L}}$ in place of (13):

$$Y_{n+1}^i = Y_n^i + \tfrac{1}{\sqrt{L}} P_{Y_n^i}^\perp \left[ V_{n+1} \tilde{\mathsf{A}}_\beta(Y_n^i, \boldsymbol{Y}_n) \right] + \tfrac{1}{L} G(Y_n^i, V_{n+1} \tilde{\mathsf{A}}_\beta(Y_n^i, \boldsymbol{Y}_n)), \quad i \in \{1, \ldots, N\}, \tag{14}$$

and we assume that $\boldsymbol{Y}_0 = \boldsymbol{X}_0$. Note that, for matrices $\{V_n\}$ with uniformly bounded entries, the functional families $\{G(X, V_n \tilde{\mathsf{A}}_\beta(X, \boldsymbol{X}))\}_{n=0}^\infty$ and $\{P_X^\perp [V_n \tilde{\mathsf{A}}_\beta(X, \boldsymbol{X})]\}_{n=0}^\infty$ are uniformly Lipschitz[5], with Lipschitz constants $M_G$ and $M_P$, respectively. Then, a standard Grönwall-type argument implies that, for any fixed time horizon $T$,

$$\max_{1 \le i \le N} \max_{1 \le k \le \lfloor TL \rfloor} \mathbb{E} \left\| X_k^i - Y_k^i \right\|^2 \to 0 \quad \text{as } L \to \infty. \tag{15}$$

We carry out this argument below. Indeed, consider the difference $\Delta_n^i := X_n^i - Y_n^i$:

$$\Delta_{n+1}^i = \Delta_n^i + \tfrac{1}{\sqrt{L}} \left( P_{X_n^i}^\perp \left[ V_{n+1} \tilde{\mathsf{A}}_\beta(X_n^i, \boldsymbol{X}_n) \right] - P_{Y_n^i}^\perp \left[ V_{n+1} \tilde{\mathsf{A}}_\beta(Y_n^i, \boldsymbol{Y}_n) \right] \right) +$$
$$+ \tfrac{1}{L} \left( G(X_n^i, V_{n+1} \tilde{\mathsf{A}}_\beta(X_n^i, \boldsymbol{X}_n)) - G(Y_n^i, V_{n+1} \tilde{\mathsf{A}}_\beta(Y_n^i, \boldsymbol{Y}_n)) \right) + \tfrac{1}{L^{3/2}} R_n^i,$$

where $R_n^i = R(X_n^i, V_{n+1} \tilde{\mathsf{A}}_\beta(X_n^i, \boldsymbol{X}_n))$. Moreover, we have

$$\Delta_{n+1}^i = \tfrac{1}{\sqrt{L}} \sum_{k=1}^n \left( P_{X_k^i}^\perp \left[ V_{k+1} \tilde{\mathsf{A}}_\beta(X_k^i, \boldsymbol{X}_k) \right] - P_{Y_k^i}^\perp \left[ V_{k+1} \tilde{\mathsf{A}}_\beta(Y_k^i, \boldsymbol{Y}_k) \right] \right) +$$
$$+ \tfrac{1}{L} \sum_{k=1}^n \left( G(X_k^i, V_{k+1} \tilde{\mathsf{A}}_\beta(X_k^i, \boldsymbol{X}_k)) - G(Y_k^i, V_{k+1} \tilde{\mathsf{A}}_\beta(Y_k^i, \boldsymbol{Y}_k)) \right) + \tfrac{1}{L^{3/2}} \sum_{k=1}^n R_k^i.$$

We also define the difference between all tokens as $\Delta_n := \boldsymbol{X}_n - \boldsymbol{Y}_n$. Note that, for the Euclidean norms in $\mathbb{R}^{d \times N}$ and $\mathbb{R}^d$, we have the following equality

$$\| \Delta_n \|^2 = \sum_{k=1}^N \| \Delta_n^k \|^2. \tag{16}$$

Now, applying the Cauchy–Schwarz inequality and using independence of the entries of $\{V_n\}$ for different values of $n$ (discrete Itô isometry), we obtain

$$\mathbb{E} \| \Delta_{n+1}^i \|^2 \le 3 \left( \tfrac{M_P^2}{L} + \tfrac{M_G^2}{L} \right) \sum_{k=1}^n \mathbb{E} \| \Delta_k \|^2 + \tfrac{3R^2}{L^2},$$

where $R = \max_{i,n} R_n^i$. Summing the inequality above and applying the property of the norm (16), we obtain

$$\mathbb{E} \| \Delta_{n+1} \|^2 \le 3N \left( \tfrac{M_P^2}{L} + \tfrac{M_G^2}{L} \right) \sum_{k=1}^n \mathbb{E} \| \Delta_k \|^2 + \tfrac{3R^2 N}{L^2}.$$

Thus, by discrete Grönwall inequality we have $\mathbb{E} \| \Delta_{n+1} \|^2 \le \frac{3R^2 N}{L^2} \exp\left(3N(M_P^2 + M_G^2)T\right)$ for $n+1 \le \lfloor LT \rfloor$. Since $N$, $R$, $T$, $M_P$, and $M_G$ are treated as fixed constants, this yields the convergence (15).

---

[5]That is, all functions in the family are $M$-Lipschitz for a common constant $M \in \mathbb{R}_{\ge 0}$.

Clearly, the same uniform approximation result holds for the linear interpolations of the sequences $\{X_n^i\}_{n \in \mathbb{N}_0} \subset \mathbb{R}^d$ and $\{Y_n^i\}_{n \in \mathbb{N}_0} \subset \mathbb{R}^d$, denoted respectively by $\boldsymbol{X}_L(t)$ and $\boldsymbol{Y}_L(t)$:

$$\max_{1 \leq i \leq N} \sup_{0 \leq t \leq T} \mathbb{E}\|X_L^i(t) - Y_L^i(t)\|^2 \to 0 \quad \text{as } L \to \infty. \tag{17}$$

We now proceed to the second step.

**Step 2. [Weak Convergence]** This step crucially relies on the work of Watanabe (1984a), which establishes convergence in terms of the martingale problem together with the uniqueness of the solution. Watanabe's theorem reduces the proof to identifying the limiting process generator via the first two conditional moments of the one-step increment.

Indeed, according to Theorem 1 therein, we can explicitly compute the coefficients of the generator $R$ corresponding to the solution of the limiting martingale problem $\mathrm{MP}(R, \mathrm{Law}(\boldsymbol{X}_0))$.

Assumptions I)–VI) of Watanabe (1984a) are satisfied under our setting for the following reasons. First, all coefficient functions involved in the scheme are infinitely differentiable; in particular, they are more regular than required, since the theorem only assumes a finite number of bounded derivatives. Second, the driving random matrices $\{V_n\}_{n \geq 1}$ form an i.i.d. sequence, which is stronger than the stationarity assumptions imposed in the general framework. Third, all derivatives of the coefficients are locally bounded, a direct consequence of the smoothness of $\tilde{\mathsf{A}}_\beta$ together with the compactness of its support. Finally, the mean drift condition holds trivially, since the noise has zero mean due to the independence and centering of $\{V_n\}$, see Assumption 1.

The expectations in Assumption VII) could be computed explicitly using the following identities:

$$\mathbb{E}\langle x, Vy \rangle^2 = \sigma^2 \cdot \|x\|^2 \cdot \|y\|^2, \quad \mathbb{E}\langle Vy, Vy \rangle = \sigma^2 \cdot d \cdot \|y\|^2, \quad \mathbb{E}\langle x, Vy \rangle Vy = \sigma^2 \cdot \|y\|^2 \cdot x \in \mathbb{R}^d. \tag{18}$$

Treating the entire sequence of the tokens as a vector in $(\mathbb{R}^d)^N$, we get that

$$c(\boldsymbol{X}) = \sigma^2 \frac{1-d}{2} \begin{bmatrix} \|\tilde{\mathsf{A}}_\beta(X^1, \boldsymbol{X})\|^2 X^1 \\ \vdots \\ \|\tilde{\mathsf{A}}_\beta(X^N, \boldsymbol{X})\|^2 X^N \end{bmatrix} \in (\mathbb{R}^d)^N, \quad b_{k\ell}(\boldsymbol{X}) = 0,$$

Computing $A(\boldsymbol{X}) = (a_{k\ell}(\boldsymbol{X}))_{1 \leq k, \ell \leq d \cdot N}$ is more delicate, because we have a diffusion in $(\mathbb{R}^d)^N$:

$$A(\boldsymbol{X}) = \mathbb{E} \underbrace{\begin{bmatrix} P_{X^1}^\perp \left[ V\tilde{\mathsf{A}}_\beta(X^1, \boldsymbol{X}) \right] \\ \vdots \\ P_{X^N}^\perp \left[ V\tilde{\mathsf{A}}_\beta(X^N, \boldsymbol{X}) \right] \end{bmatrix}}_{\in \mathbb{R}^{dN}} \left[ \left[ P_{X^1}^\perp \left[ V\tilde{\mathsf{A}}_\beta(X^1, \boldsymbol{X}) \right] \right]^T \quad \cdots \quad \left[ P_{X^N}^\perp \left[ V\tilde{\mathsf{A}}_\beta(X^N, \boldsymbol{X}) \right] \right]^T \right].$$

A direct identification shows that the generator $R$, defined by

$$R\phi(\boldsymbol{X}) = \langle c(\boldsymbol{X}), \nabla\phi(\boldsymbol{X}) \rangle + \tfrac{1}{2} \mathrm{Tr}\left[ A(\boldsymbol{X})\nabla^2\phi(\boldsymbol{X}) \right],$$

coincides with the generator $R'$ associated with the following system:

$$dY_t^i = \frac{(1-d)\sigma^2}{2} \|\tilde{\mathsf{A}}_\beta(Y_t^i, \boldsymbol{Y}_t)\|^2 Y_t^i \, dt + \sigma \, P_{Y_t^i}^\perp \left[ dW_t \, \tilde{\mathsf{A}}_\beta(Y_t^i, \boldsymbol{Y}_t) \right], \quad \boldsymbol{Y}_0 = \boldsymbol{X}_0, \qquad i = 1, \dots, N, \tag{19}$$

treated as an $(\mathbb{R}^d)^N$ valued diffusion process. Moreover, we enjoy the fact that $\mathrm{MP}(R', \mathrm{Law}(\boldsymbol{Y}_0))$ is a weak solution of (19), see for instance Theorem 1 in Kurtz (2010) (local boundedness and measurability requirements are satisfied). As a bonus, we get that this weak solution is distributionally unique, because the solution of the martingale problem is unique.

By a standard argument, which we reproduce in Lemma 1, any solution of (19) remains almost surely supported on the unit sphere. Consequently, along such trajectories one has $\tilde{\mathsf{A}}_\beta = \mathsf{A}_\beta$ in (19). This completes the identification of the limit and shows that the linearized dynamics converges to (6).

**Step 3. [Passage to the Limit]** We apply a Slutsky-type argument; see Theorem 6b) in Ferguson (2017), which extends without modification to Polish spaces. Indeed, note that

1. convergence (17) implies convergence in probability: $\boldsymbol{X}_L - \boldsymbol{Y}_L \xrightarrow{\mathbb{P}} 0$;

2. according to Step 2, we have $\boldsymbol{Y}_L \xrightarrow{d} \boldsymbol{Y}$.

It follows that $(\boldsymbol{X}_L - \boldsymbol{Y}_L) + \boldsymbol{Y}_L \xrightarrow{d} \boldsymbol{Y}$, which completes the proof of the theorem. $\qquad\square$

*Proof of Lemma 1.* The result can be easily deduced by applying Itô's formula to $\|Y_t\|^2 = \langle Y_t, Y_t \rangle$. We immediately obtain

$$d\langle Y_t^i, Y_t^i \rangle = 2\langle Y_t^i, dY_t^i \rangle + \langle dY_t^i, dY_t^i \rangle.$$

Using the fact that $\langle P_x^\perp[y], x \rangle = 0$ for all $y \in \mathbb{R}^d$ and performing some simplifications, we obtain

$$d\|Y_t^i\|^2 = \sigma^2 \|\mathsf{A}_\beta(Y_t^i, \boldsymbol{Y}_t)\|^2 (\|Y_t^i\|^2 - d)(\|Y_t^i\|^2 - 1)\, dt,$$

which implies that $Y_{t+s}^i \in \mathbb{S}^{d-1}$ for all $s > 0$ as long as $Y_t^i \in \mathbb{S}^{d-1}$. Since we have the initial condition $\|Y_0^i\| = 1$, we immediately obtain the desired statement. $\qquad\square$

*Proof of Lemma 2.* Similarly to the first step of the proof of Theorem 1, for $x \in \mathbb{S}^{d-1}$ and $y \in \mathbb{R}^d$ we consider the following function:

$$F_\delta(x, y) = (x + \delta y)/\|x + \delta y\| = \frac{(x + \delta y)}{\sqrt{\langle x, x \rangle + 2\delta\langle x, y \rangle + \delta^2\langle y, y \rangle}}.$$

Trivially, we obtain

$$F_\delta(x, y) = (x + \delta y)\left(1 + 2\delta A(x, y) + \delta^2 B(x, y)\right)^{-1/2},$$

where we denote

$$A(x, y) := \langle x, y \rangle, \quad B(x, y) := \langle y, y \rangle.$$

By directly applying the Taylor expansion and assuming that $x \in \mathbb{S}^{d-1}$ and $y$ belongs to a compact subset $\mathcal{K} \subset \mathbb{R}^d$, we obtain a more convenient form for sufficiently small $\delta > 0$:

$$F_\delta(x, y) = x + \delta P_x^\perp[y] + \delta^2 \left(\frac{3}{2}A^2 x - \frac{1}{2}Bx - Ay\right) + \delta^3 R(x, y),$$

where $\|R(x, y)\| \leq C$. We note that in our case, $\delta = \frac{1}{L}$. This corresponds to a deterministic scaling regime, in contrast with the diffusive scaling $\delta = L^{-1/2}$ used in Theorem 1. Also, note that we are working on the product of the unit spheres. Thus, the remainder of order $1/L^2$ is bounded. This means that for every token number $i \leq N$, we have

$$X_{n+1}^i - X_n^i = \frac{1}{L}P_{X_n^i}^\perp\left[\bar{V}\mathsf{A}_\beta(X_n^i, \boldsymbol{X}_n)\right] + \bar{o}\left(\frac{1}{L}\right),$$

where the remainder converges to zero uniformly for all $n \leq L$. A standard Grönwall argument implies the uniform convergence to the solution of the following problem:

$$\begin{cases} \frac{dY_t^i}{dt} = P_{Y_t^i}^\perp\left[\bar{V}\mathsf{A}_\beta(Y_t^i, \boldsymbol{Y}_t)\right], & \text{for } i \leq N, \\ \boldsymbol{Y}_0 = \boldsymbol{X}_0. \end{cases}$$

Thus, we have obtained the required statement. $\qquad\square$

*Proof of Lemma 3.* We note that for the dynamics (7), we apply exactly the same technique as for Theorem 1. In this case, instead of the truncated dynamics (14), we obtain the following dynamics:

$$Y_{n+1}^i = Y_n^i + \frac{\varepsilon v_{n+1}}{\sqrt{L}} P_{Y_n^i}^\perp\left[\tilde{\mathsf{A}}_\beta(Y_n^i, \boldsymbol{Y}_n)\right] + \frac{1}{L}\left[G(Y_n^i, \varepsilon v_{n+1}\tilde{\mathsf{A}}_\beta(Y_n^i, \boldsymbol{Y}_n)) + P_{Y_n^i}^\perp[\bar{V}\tilde{\mathsf{A}}_\beta(Y_n^i, \boldsymbol{Y}_n)]\right], \quad i \in \{1, \dots, N\}.$$

Then, for the linearized dynamics above and the recurrence relation (7), we apply exactly the same argument. This leads to the result of the lemma. $\qquad\square$

# C. Long-Time Behavior and Clustering

**Outline of the Analysis.** This section analyzes the long-time behavior of the overlap dynamics underlying clustering. We first derive a closed one-dimensional description of the overlap process and classify its boundary points using the scale–speed framework of Karlin & Taylor (1981) (Lemmas 4, 5, and 6). Based on this classification, we then prove the clustering and anti-clustering results by exploiting the martingale properties of the associated scale function, making explicit a classical implication of one-dimensional diffusion theory in our setting.

## C.1. Overlap Dynamics and Reduction to One Dimension

We begin by observing that clustering events admit a simple geometric characterization in terms of pairwise inner products. Let $(Y_t^i)_{i=1}^N \subset \mathbb{S}^{d-1}$ be token trajectories solving the Transformer SDE (6). For any pair $(i, j)$, the squared distance and squared sum satisfy

$$\|Y_t^i - Y_t^j\|^2 = 2\big(1 - \langle Y_t^i, Y_t^j \rangle\big), \qquad \|Y_t^i + Y_t^j\|^2 = 2\big(1 + \langle Y_t^i, Y_t^j \rangle\big),$$

since the tokens remain on the unit sphere. Consequently,

$$\|Y_t^i - Y_t^j\| \longrightarrow 0 \quad \Longleftrightarrow \quad \langle Y_t^i, Y_t^j \rangle \longrightarrow 1,$$

which corresponds to clustering, while

$$\|Y_t^i + Y_t^j\| \longrightarrow 0 \quad \Longleftrightarrow \quad \langle Y_t^i, Y_t^j \rangle \longrightarrow -1,$$

corresponding to convergence toward an antipodal configuration. Thus, clustering and anti-clustering events are entirely characterized by the long-time behavior of the overlap process

$$Z_t^{i,j} := \langle Y_t^i, Y_t^j \rangle \in [-1, 1].$$

For general $N$, applying Itô's formula to $Z_t^{i,j}$ yields a one-dimensional stochastic differential equation whose coefficients depend on the remaining tokens through the attention weights. Hence the overlap dynamics are not closed in general. The situation becomes particularly tractable in the case $N = 2$, where the overlap forms a closed Markov diffusion.

**Lemma 4** (Closed overlap dynamics for two tokens: unnormalized vs. normalized self-attention)**.** *Assume $N = 2$ and $K = Q = \mathrm{Id}(d)$. Let $(Y_t^1, Y_t^2) \in (\mathbb{S}^{d-1})^2$ solve (6) and define the overlap*

$$Z_t := \langle Y_t^1, Y_t^2 \rangle \in [-1, 1].$$

*Then $(Z_t)_{t \geq 0}$ is a one-dimensional diffusion. More precisely:*

*(i) **Unnormalized self-attention** $\mathsf{A}_\beta = \mathsf{A}_\beta^{\mathsf{U}}$. There exists a scalar Brownian motion $B_t$ such that*

$$dZ_t = \tfrac{\sigma^2}{4}(1 - Z_t^2)\left(2(d-2)e^{\beta(1+Z_t)} - Z_t(e^{2\beta} + e^{2\beta Z_t})\right)dt + \frac{\sigma}{\sqrt{2}}(1 - Z_t^2)\sqrt{e^{2\beta} + e^{2\beta Z_t}}\,dB_t. \tag{20}$$

*(ii) **Normalized self-attention** $\mathsf{A}_\beta = \mathsf{A}_\beta^{\mathsf{S}}$. There exists a scalar Brownian motion $\widetilde{B}_t$ such that*

$$dZ_t = \sigma^2 \frac{(1 - Z_t^2)}{(e^\beta + e^{\beta Z_t})^2}\left(2(d-2)e^{\beta(1+Z_t)} - Z_t(e^{2\beta} + e^{2\beta Z_t})\right)dt + \sigma \frac{(1 - Z_t^2)}{e^\beta + e^{\beta Z_t}}\sqrt{e^{2\beta} + e^{2\beta Z_t}}\,d\widetilde{B}_t. \tag{21}$$

*Proof.* We write the overlap as

$$Z_t = \langle Y_t^1, Y_t^2 \rangle,$$

and apply Itô's formula to the bilinear map $(x, y) \mapsto \langle x, y \rangle$. We obtain the identity

$$dZ_t = \langle dY_t^1, Y_t^2 \rangle + \langle Y_t^1, dY_t^2 \rangle + \langle dY_t^1, dY_t^2 \rangle. \tag{22}$$

We now substitute the dynamics of $(Y_t^1, Y_t^2)$ from (6). We recall that

$$P_x^\perp [dW_t f(x)] = \sum_{k,l=1}^d c_{kl}(x)dW_t^{kl}, \quad c_{kl}(x) = f_l(x)(e_k - x_k x) \in \mathbb{R}^d,$$

where $e_k := \begin{bmatrix} 0 & \dots & \underbrace{1}_{k} & \dots & 0 \end{bmatrix}^T$ and $f \in \mathbb{R}^d$. Expanding the expressions similarly in (22) and using pairwise independence of entries, we get:

$$dZ_t = \sigma^2 \left( \tfrac{1}{2}(1-d)Z_t \left( \|\mathsf{A}_\beta(Y_t^1, \boldsymbol{Y}_t)\|^2 + \|\mathsf{A}_\beta(Y_t^2, \boldsymbol{Y}_t)\|^2 \right) + (Z_t^2 + d - 2)\left\langle \mathsf{A}_\beta(Y_t^1, \boldsymbol{Y}_t), \mathsf{A}_\beta(Y_t^2, \boldsymbol{Y}_t) \right\rangle \right) dt +$$
$$+ \sigma \sum_{k,l=1}^{d} \left[ Y_t^{2,k} \mathsf{A}_\beta^l(Y_t^1, \boldsymbol{Y}_t) + Y_t^{1,k} \mathsf{A}_\beta^l(Y_t^2, \boldsymbol{Y}_t) - Z_t \left( Y_t^{1,k} \mathsf{A}_\beta^l(Y_t^1, \boldsymbol{Y}_t) + Y_t^{2,k} \mathsf{A}_\beta^l(Y_t^2, \boldsymbol{Y}_t) \right) \right] dW_t^{kl},$$

where $\mathsf{A}_\beta^k(Y, \boldsymbol{Y}) := \langle \mathsf{A}_\beta(Y, \boldsymbol{Y}), e_k \rangle$ and $Y_t^{i,k} := \langle Y_t^i, e_k \rangle$. Normalizing the diffusion component, there exists a scalar Brownian motion $B_t \in \mathbb{R}$ (correlated with the original ones by the quadratic covariation property) such that:

$$\sum_{k,l=1}^{d} \left[ Y_t^{2,k} \mathsf{A}_\beta^l(Y_t^1, \boldsymbol{Y}_t) + Y_t^{1,k} \mathsf{A}_\beta^l(Y_t^2, \boldsymbol{Y}_t) - Z_t \left( Y_t^{1,k} \mathsf{A}_\beta^l(Y_t^1, \boldsymbol{Y}_t) + Y_t^{2,k} \mathsf{A}_\beta^l(Y_t^2, \boldsymbol{Y}_t) \right) \right] dW_t^{kl}$$
$$= \sqrt{(1-Z_t^2)\left(\|\mathsf{A}_\beta(Y_t^1, \boldsymbol{Y}_t)\|^2 + \|\mathsf{A}_\beta(Y_t^2, \boldsymbol{Y}_t)\|^2 - 2Z_t\langle \mathsf{A}_\beta(Y_t^1, \boldsymbol{Y}_t), \mathsf{A}_\beta(Y_t^2, \boldsymbol{Y}_t)\rangle\right)}\, dB_t. \quad (23)$$

Thus, the final expression for arbitrary two tokens is:

$$dZ_t = \sigma^2 \left( \tfrac{1}{2}(1-d)Z_t \left( \|\mathsf{A}_\beta(Y_t^1, \boldsymbol{Y}_t)\|^2 + \|\mathsf{A}_\beta(Y_t^2, \boldsymbol{Y}_t)\|^2 \right) + (Z_t^2 + d - 2)\left\langle \mathsf{A}_\beta(Y_t^1, \boldsymbol{Y}_t), \mathsf{A}_\beta(Y_t^2, \boldsymbol{Y}_t) \right\rangle \right) dt +$$
$$+ \sigma \sqrt{(1-Z_t^2)\left(\|\mathsf{A}_\beta(Y_t^1, \boldsymbol{Y}_t)\|^2 + \|\mathsf{A}_\beta(Y_t^2, \boldsymbol{Y}_t)\|^2 - 2Z_t\langle \mathsf{A}_\beta(Y_t^1, \boldsymbol{Y}_t), \mathsf{A}_\beta(Y_t^2, \boldsymbol{Y}_t)\rangle\right)}\, dB_t.$$

For $N = 2$, we substitute the explicit formulas for unnormalized and normalized attention. For unnormalized attention $\mathsf{A}_\beta^{\mathrm{U}}$, we have

$$\mathsf{A}_\beta^{\mathrm{U}}(Y^i, \boldsymbol{Y}) = \frac{1}{2}\left(Y^i e^\beta + Y^{3-i} e^{\beta Z_t}\right), \quad i = 1, 2.$$

For normalized attention $\mathsf{A}_\beta^{\mathrm{S}}$, we have

$$\mathsf{A}_\beta^{\mathrm{S}}(Y^i, \boldsymbol{Y}) = \frac{Y^i e^\beta + Y^{3-i} e^{\beta Z_t}}{e^\beta + e^{\beta Z_t}}, \quad i = 1, 2.$$

Substituting these expressions and simplifying algebraically yields exactly (20) for case (i) and (21) for case (ii), as claimed. □

## C.2. Boundary Behavior of the Overlap Diffusion

By Lemma 4, the overlap process $(Z_t)_{t \geq 0}$ is a one-dimensional Itô diffusion on $[-1, 1]$ with smooth coefficients on $(-1, 1)$ that vanish only at the boundaries $\{-1, 1\}$; the diffusion coefficient is strictly positive in the interior. Hence, the long-time behavior is determined entirely by whether $Z_t$ can converge to $\pm 1$.

We employ Feller's boundary classification for one-dimensional diffusions (Karlin & Taylor, 1981, Chapter 15, §6, p.226). Both boundaries $\pm 1$ are inaccessible, but their attracting/non-attracting character depends on $(d, \beta)$. The analysis reveals that, for each $(d, \beta)$ and attention mechanism, the boundaries can become attracting. As a result, $Z_t$ converges to either $+1$ or $-1$ with positive probability, with no other possible limits.

**Lemma 5** (Boundary classification for two tokens, full noise). *Consider the overlap diffusion $Z_t$ for $N = 2$ tokens under the full noise model, for either unnormalized self-attention $\mathsf{A}_\beta^{\mathrm{U}}$ (SDE (20)) or normalized self-attention $\mathsf{A}_\beta^{\mathrm{S}}$ (SDE (21)). The boundary behavior is identical for both attention mechanisms and is given as follows:*

1. *For dimension $d = 2$, both boundaries $+1$ and $-1$ are attracting.*

2. *For dimensions $d \geq 3$:*
   (a) *The boundary $+1$ is always attracting.*
   (b) *The boundary $-1$ is attracting if and only if $d - 2 < \cosh(2\beta)$. Otherwise, it is non-attracting.*

*Proof.* Both SDEs can be written in the standard form

$$dZ_t = \mu(Z_t)dt + \sigma(Z_t)dB_t,$$

with drift and diffusion coefficients that are smooth on $(-1, 1)$ and vanish at the endpoints. Crucially, the ratio $\frac{2\mu(z)}{\sigma^2(z)}$—which determines the scale function—is identical for the two attention mechanisms. Indeed,

$$\frac{2\mu(z)}{\sigma^2(z)} = \frac{2(d-2)e^{\beta(1+z)} - z(e^{2\beta} + e^{2\beta z})}{(1-z^2)(e^{2\beta} + e^{2\beta z})},$$

as can be verified directly from (20) and (21). Therefore, the derivative of the scale function (with base point $x_0 \in (-1, 1)$):

$$\boldsymbol{s}(x) = \exp\left[-\int_{x_0}^{x} \frac{2\mu(y)}{\sigma^2(y)}\, dy\right] = \exp\left[-\int_{x_0}^{x} \frac{2(d-2)e^{\beta(1+y)} - y(e^{2\beta} + e^{2\beta y})}{(1-y^2)(e^{2\beta} + e^{2\beta y})}\, dy\right],$$

is the same for both dynamics.

Define $\psi(y) := \dfrac{2(d-2)e^{\beta(1+y)}}{e^{2\beta} + e^{2\beta y}}$. Then

$$\boldsymbol{s}(x) = \begin{cases} \sqrt{\dfrac{1-x_0^2}{1-x^2}}, & d = 2, \\[3ex] \sqrt{\dfrac{1-x_0^2}{1-x^2}} \exp\left[-\displaystyle\int_{x_0}^{x} \frac{\psi(y)}{1-y^2}\, dy\right], & d \geq 3. \end{cases}$$

For $d \geq 3$, near the right boundary we have $\psi(1) = d - 2$ and

$$\int_{x_0}^{1-\varepsilon} \frac{\psi(y)}{1-y}\, dy = \frac{d-2}{2}\left|\log\varepsilon\right| + O(1), \qquad \varepsilon \to 0^+.$$

Hence $\boldsymbol{s}(1 - \varepsilon) \asymp \varepsilon^{-\frac{1}{2} + \frac{d-2}{2}}$. The integral $\int_{x_0}^{1} \boldsymbol{s}(x)dx$ converges for all $d \geq 2$, so the boundary $+1$ is always attracting.

Near the left boundary, $\psi(-1) = \dfrac{d-2}{\cosh(2\beta)}$ and

$$\int_{-1+\varepsilon}^{x_0} \frac{\psi(y)}{1+y}\, dy = \frac{d-2}{2\cosh(2\beta)}\left|\log\varepsilon\right| + O(1), \qquad \varepsilon \to 0^+.$$

Thus $\boldsymbol{s}(-1 + \varepsilon) \asymp \varepsilon^{-\frac{1}{2} - \frac{d-2}{2\cosh(2\beta)}}$. The boundary $-1$ is attracting precisely when $\int_{-1}^{x_0} \boldsymbol{s}(x)dx < \infty$, which occurs iff $-\frac{1}{2} - \frac{d-2}{2\cosh(2\beta)} > -1$, i.e. $d - 2 < \cosh(2\beta)$.

For $d = 2$, the explicit scale function yields $\boldsymbol{s}(1-\varepsilon) \asymp \varepsilon^{-1/2}$ and $\boldsymbol{s}(-1+\varepsilon) \asymp \varepsilon^{-1/2}$, so both boundaries are attracting. $\square$

**Lemma 6** (Attainability). *For the diffusion processes defined in (20), (21), (26), with $Z_0 \in (-1, 1)$, we have the following properties:*

1. *$|Z_t| \leq 1$ a.s. for all $t \geq 0$;*

2. *Both boundaries $\pm 1$ are inaccessible; that is, $|Z_t| < 1$ for any finite $t$.*

*Proof.* The first item follows directly from the definition $Z_t = \langle X_t^1, X_t^2 \rangle$ for two unit vectors $X_t^1, X_t^2$ and the Cauchy–Schwarz inequality. For the second item, we could verify it directly by computing the corresponding integral $\Sigma$ in Karlin & Taylor (1981), but instead we consider the function $f(x) = \tanh^{-1}(x)$, defined as

$$f(x) := \tfrac{1}{2}\ln\left(\frac{1+x}{1-x}\right).$$

For the generic dynamics

$$dZ_t = (1 - Z_t^2)(c(Z_t)\,dt + \gamma(Z_t)\,dB_t),$$

which correspond to (20), (21), (26), where $c$ and $\gamma$ are smooth and bounded on $[-1, 1]$, we consider $Y_t = f(Z_t)$. Applying Itô's formula, we obtain:

$$dY_t = \left(c \circ \tanh(Y_t) + Z_t \cdot (\gamma \circ \tanh(Y_t))^2\right) dt + \gamma \circ \tanh(Y_t)\,dB_t, \quad Y_0 = f(Z_0).$$

By Itô's Uniqueness and Existence Theorem, $Y_t$ is well-defined and does not explode to $\pm\infty$ in finite time. Thus, $Z_t = \tanh(Y_t)$ never reaches $\pm 1$. $\qquad\square$

## C.3. Proof of the Phase Transition Result (Theorem 2)

With the boundary classification of the overlap diffusion established in Lemmas 5,6, we are now equipped to prove the main result of this section. Before doing so, we introduce some additional notation. Based on the function $s(x)$, we define a *scale function*:

$$\boldsymbol{S}(x) = \int_{x_0}^{x} \boldsymbol{s}(y)\,dy.$$

The main property is that directly applying Itô's formula, we get that $\boldsymbol{S}(Z_t)$ is a local martingale. Thanks to the classification of the boundary points, we know precisely the range of parameters $(\beta, d)$ for which $\boldsymbol{S}(\pm 1)$, defined as the corresponding limit, is finite, and vice versa. Note that this behavior does not depend on the choice of the initial point $x_0 \in (-1, 1)$. We then define a convenient set of stopping times:

$$\tau_a := \inf\{t \geqslant 0 \mid Z_t = a\} \text{ for } a \in (-1, 1).$$

Although Theorem 2 follows directly from the boundary classification above, the limiting behavior itself requires a proof. While this result is folklore and appears in various forms across different textbooks, we have struggled to find it stated in exactly the form we need. For the sake of clarity, we therefore include a proof based on unpublished lecture notes by Jacobsen (2008).

*Proof of Theorem 2.* For the case when two initial token positions do not coincide, we have $Z_0 = z_0 \in (-1, 1)$. We restrict ourselves to the case when $z_0$ is non-random, but the results remain the same for random initial distributions by convolution. We note that both boundaries $\pm 1$ are unattainable in finite time by Lemma 6.

The proof relies on the properties of the function $\boldsymbol{S}(x)$ defined above. If $\boldsymbol{S}(1) < \infty$, then

$$\lim_{t\to\infty} Z_t = 1 \quad \mathbb{P}^{z_0}\text{-almost surely on } A_-, \quad \text{where } A_- = \bigcup_{a:\, a < z_0} \{\tau_a = \infty\}, \quad \mathbb{P}^{z_0}(A_-) = \frac{\boldsymbol{S}(z_0) - \boldsymbol{S}(-1)}{\boldsymbol{S}(1) - \boldsymbol{S}(-1)}. \quad (24)$$

Moreover, if $\boldsymbol{S}(-1) > -\infty$, then

$$\lim_{t\to\infty} Z_t = -1 \quad \mathbb{P}^{z_0}\text{-almost surely on } A_+, \quad \text{where } A_+ = \bigcup_{b:\, b > z_0} \{\tau_b = \infty\}, \quad \mathbb{P}^{z_0}(A_+) = \frac{\boldsymbol{S}(1) - \boldsymbol{S}(z_0)}{\boldsymbol{S}(1) - \boldsymbol{S}(-1)}.$$

Indeed, let's show that for $\boldsymbol{S}(1) < \infty$, we have (24). Clearly, for $a < z_0$ $\{\tau_a = \infty\} \uparrow A_-$ as $a \to -1$, then

$$\mathbb{P}^{z_0}(A_-) = \lim_{a\downarrow -1} \mathbb{P}^{z_0}(\tau_a = \infty).$$

Note that $\boldsymbol{S}(Z_{t\wedge\tau_a}) := \boldsymbol{S}(Z_{\min\{t,\tau_a\}})$ is a bounded local martingale, hence a true martingale. Boundedness follows from the fact that $\boldsymbol{S}(Z_{t\wedge\tau_a}) \in [\boldsymbol{S}(a), \boldsymbol{S}(1)]$ and we assumed that $\boldsymbol{S}(1) < \infty$. Then, by the martingale convergence theorem, we can define $\boldsymbol{S}(Z_{\tau_a})$ as follows:

$$\boldsymbol{S}(Z_{\tau_a}) := \begin{cases} \boldsymbol{S}(a) \text{ on } (\tau_a < \infty), \\ \lim_{t\to\infty} \boldsymbol{S}(Z_t) \text{ on } \tau_a = \infty. \end{cases}$$

So, we have on one hand

$$\mathbb{E}^{z_0}\left[\boldsymbol{S}(Z_{\tau_a})\right] = \boldsymbol{S}(z_0), \tag{25}$$

on the other hand, by inaccessibility of 1

$$\mathbb{P}^{z_0}\left[\tau_a = \infty\right] = \lim_{\substack{z_0 < b \\ b \to 1}} \mathbb{P}^{z_0}\left[\tau_a > \tau_b\right] = \lim_{\substack{z_0 < b \\ b \to 1}} \frac{\boldsymbol{S}(z_0) - \boldsymbol{S}(a)}{\boldsymbol{S}(b) - \boldsymbol{S}(a)} = \frac{\boldsymbol{S}(z_0) - \boldsymbol{S}(a)}{\boldsymbol{S}(1) - \boldsymbol{S}(a)},$$

Substituting this into (25), we get

$$\boldsymbol{S}(z_0) = \mathbb{E}^{z_0}\left[\boldsymbol{S}(Z_{\tau_a})\right] = \boldsymbol{S}(a) \cdot (1 - \mathbb{P}^{z_0}\left[\tau_a = \infty\right]) + \mathbb{E}^{z_0}\left[\boldsymbol{S}(Z_{\tau_a})\operatorname{Ind}\{\tau_a = \infty\}\right],$$

and thus simplifying we get that

$$\mathbb{E}^{z_0}\left[\boldsymbol{S}(Z_{\tau_a})\operatorname{Ind}\{\tau_a = \infty\}\right] = \boldsymbol{S}(1)\mathbb{P}(\tau_a = \infty).$$

Since $\boldsymbol{S}(Z_{\tau_a}) \le \boldsymbol{S}(1)$, we have that $\boldsymbol{S}(Z_{\tau_a}) = \boldsymbol{S}(1)\ \mathbb{P}^{z_0}$ almost surely on $\{\tau_a = \infty\}$. Then, (24) is proved. From this and the boundary classification in Lemma 5, we get the statement of the theorem. □

## C.4. Hybrid Model Boundary Behavior

We now turn to the hybrid clustering regime.

*Proof of Lemma 1.* The overlap equation similar to (21) and (20) is given as follows:

$$dZ_t = e^{\beta Z_t}(Z_t^2 - 1)\left[\left(\tfrac{\varepsilon^2 e^\beta}{2} + Z_t\varepsilon^2 e^{\beta Z_t} - 1\right)dt - \varepsilon dB_t\right]. \tag{26}$$

The reasoning about convergence is exactly the same as in the proof of Theorem 2, since Lemma 6 applies to both. Here we only perform the boundary classification.

In this case, for $\boldsymbol{s}(x)$ we obtain the following expression:

$$\boldsymbol{s}(x) = \exp\left(-2\int_{x_0}^{x}\frac{\frac{\varepsilon^2 e^{\beta(1-y)}}{2} + y\varepsilon^2 - e^{-\beta y}}{\varepsilon^2(y^2 - 1)}\,dy\right) = \exp\left(\int_{x_0}^{x}\frac{(e^\beta - \frac{2}{\varepsilon^2})e^{-\beta y} + 2y}{1 - y^2}\,dy\right).$$

Simplifying the integral, we get:

$$\boldsymbol{s}(x) = \frac{1 - x_0^2}{1 - x^2}\exp\left((e^\beta - \tfrac{2}{\varepsilon^2})\int_{x_0}^{x}\frac{e^{-\beta y}}{1 - y^2}\,dy\right).$$

Hence,

$$\boldsymbol{s}(1 - \delta) \sim \frac{1}{\delta} \cdot \exp\left(-\tfrac{1}{2}(e^\beta - \tfrac{2}{\varepsilon^2})e^{-\beta}\log(\delta)\right), \quad \boldsymbol{s}(-1 + \delta) \sim \frac{1}{\delta} \cdot \exp\left(\tfrac{1}{2}(e^\beta - \tfrac{2}{\varepsilon^2})e^\beta \log(\delta)\right), \text{ as } \delta \to 0.$$

Simplifying, we get:

$$\boldsymbol{s}(1 - \delta) \sim \delta^{-\frac{3}{2} + \frac{e^{-\beta}}{\varepsilon^2}}, \quad \boldsymbol{s}(-1 + \delta) \sim \delta^{-1 + \frac{e^{2\beta}}{2} - \frac{e^\beta}{\varepsilon^2}}.$$

From the standard integrability conditions, we have

1. $\boxed{\frac{1}{\varepsilon^2} > \frac{e^\beta}{2}}$ — the boundary $x = 1$ is attracting, while $x = -1$ is non-attracting.

2. $\boxed{\frac{1}{\varepsilon^2} < \frac{e^\beta}{2}}$ — the boundary $x = 1$ is non-attracting, while $x = -1$ is attracting.

This completes the proof. □

# D. Additional Experimental Results

This section provides additional numerical experiments demonstrating the convergence dynamics of token clustering. We validate that the chosen time horizons are sufficiently large for clustering behavior to stabilize, and demonstrate the robustness of our theoretical findings across different parameter regimes.

### D.1. Phase Transition Analysis Across Parameter Regimes

To verify that our choice of time horizon in the main paper (Figure 3) is conservative, we present convergence curves at shorter time horizons ($T = 50$ and $T = 100$ versus $T = 500$ on Figure 3). Specifically, we track two key quantities:

1. The proportion of tokens that fail to cluster into any configuration

2. The proportion of observed clustering configurations, which stabilizes well below the maximum time horizon

We examine three representative values of the inverse temperature parameter $\beta$: the regimes $\beta \to 0$ and $\beta \to \infty$ where behavioral differences are most pronounced, and an intermediate regime. These regimes are chosen to demonstrate that qualitative features persist across the parameter space. *All other experimental settings are identical to those used in the main phase diagram (Figure 3), with the sole difference being the reduced time horizon. Reported curves are obtained by averaging over $2 \cdot 10^3$ independent trajectories.*

We focus on three representative regimes of the inverse temperature parameter $\beta$, corresponding to qualitatively distinct clustering behaviors. In the first regime, two-token clustering is possible for all dimensions. In the second regime, such clustering occurs only for a subset of dimensions. In the third regime, two-token clustering again becomes possible uniformly across all dimensions. The selected values of $\beta$ are chosen to illustrate these three regimes.

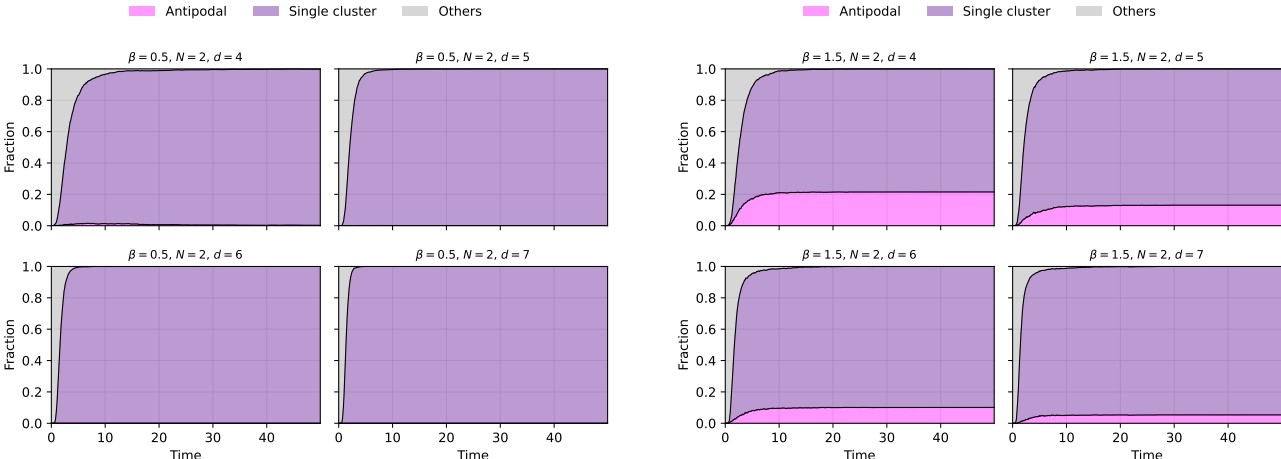

*(a)* Below-threshold regime ($\beta < 1$): convergence toward antipodal clustering does not occur for all dimensions.

*(b)* Above-threshold regime ($\beta > 1$): convergence toward antipodal clustering occurs for all dimensions.

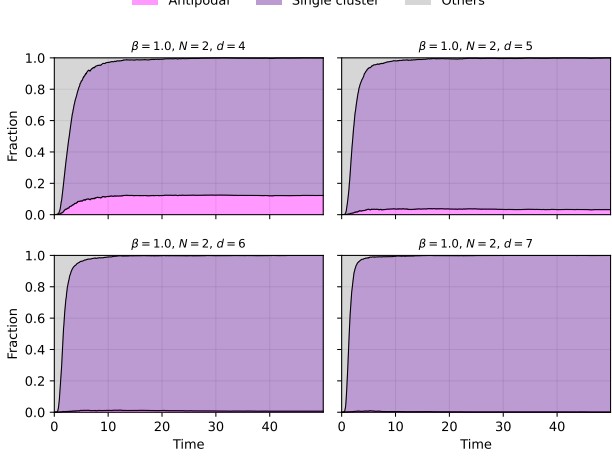

*(c)* Near-threshold regime ($\beta = 1$). Convergence to the antipodal configuration occurs only for $d < 6$.

*Figure 10.* Convergence behavior across representative values of the inverse temperature $\beta$. Panels (a)–(c) correspond respectively to regimes in which all trajectories converge to a single-cluster configuration, in which convergence to an antipodal configuration is attainable for all dimensions considered, and in which the antipodal configuration is attainable only for certain dimensions while remaining inaccessible for others.

