# OpenReview forum: "Clustering in Deep Stochastic Transformers"
_ICML.cc/2026/Conference — ICML 2026 spotlight_

### Official Review · Reviewer_24Mv · 2026-02-19

**Soundness:** 3
**Presentation:** 3
**Significance:** 2
**Originality:** 3
**Overall Recommendation:** 3
**Confidence:** 3

**Summary:**

This paper analyzes the behavior of a simplified Transformer-like model (with only self-attention layers, normalizations and residual connections), in the random initialization and infinite-depth limit. It derives an explicit SDE description of the convergent behavior. Moreover, it also performs a detailed analysis in the case of two tokens, highlighting a phase transition behavior governed by the hidden dimension and softmax rescaling factor, where the representation of the two tokens can converge to a single point or to antipodal configurations.

**Compliance With Llm Reviewing Policy:**

Affirmed.

**Key Questions For Authors:**

- In Section 4, do you need any assumption for $Q,K$ matrices? Are they set to be deterministic or random? It feels strange that only the randomness of $V$ is considered....
- Could you discuss some intuition behind the main results? For example, in Theorem 1, why does the noise term only happen in the space orthogonal to $Y_i$?

**Strengths And Weaknesses:**

Overall, this paper presents a self-contained theoretical result. The use of mathematical language is precise and clear, which makes it delightful to read this paper, and the theoretical results appear to be sound.

However, it also has significant issues. One obvious issue is that the model used in this paper is far from practical: it only considers random initialization, infinite-depth and a simplified model without feed-forward layers. I understand that many theoretical analysis relies on strong assumptions, but usually these assumptions should either be reasonable simplifications (e.g. do not affect the main result), or empirically verified. For this paper, the authors fail to address this. For example, in Section 4.2, the authors want to make "a first step toward understanding signal propagation after initialization", and assume $V$ is a rescaled identity matrix, but it is unclear why $V$ becomes identity matrix after initialization (e.g. are there any empirical evidence showing that $V$ is approximately identity?). Similar concerns also arise with Assumption 2.

The second issue is that, the authors fail to show what are the realistic consequences of their results. I agree that the two-token interaction analyzed in Section 5 is itself an interesting theoretical setting, but I don't see how this result (as well as results in Section 4) connects with real Transformer behaviors that have been observed in practice. It feels like this paper misses a discussion section that connects their simplified model and corresponding analysis with real transformers. Currently, it is unclear what new insights this paper can bring to the machine learning community.

As a summary, I do like the mathematical rigor of this paper, yet this paper clearly misses a part that grounds its theoretical analysis with practice.

---

> ### Author Rebuttal · Authors · 2026-03-30
>
> Dear Reviewer,
>
> We are glad that you liked the rigor of our work  and considered this paper delightful to read. We are happy to answer your questions one by one:
>
> ### Addressing Issue 1: Simplified Model Assumptions
>
>
> Analyzing even simplified Transformer still remains a challenge, and the introduction of multiple layers intensifies the complexity. Our assumptions align with recent literature [1, 2, 3], where studying constant $Q, K, V$ without MLPs has already yielded profound insights and difficulties. **We advance this line of work by introducing a source of randomness in the matrix $V$, moving a step closer to practical settings.**
>
> Regarding empirical verification: we agree on its importance, which is why **we specifically demonstrated the key role of $V$’s stochasticity at initialization using a real-world dataset** (CIFAR-10) in Section 6.
>
> > For example, in Section 4.2, the authors want to make "a first step toward understanding signal propagation after initialization", and assume $V$ is a rescaled identity matrix, but it is unclear why becomes identity matrix after initialization (e.g. are there any empirical evidence showing that is approximately identity?). Similar concerns also arise with Assumption 2.
>
> We **do not** claim that the drift during training is close to the identity matrix. In fact, the proof of Lemma 3 remains unchanged in the case where $\omega_{\varepsilon}(n,L)=\tfrac{1}{L}V+\varepsilon\tfrac{v_n}{\sqrt{L}} I_d,$ where $V$ is an arbitrary constant matrix. In the new revision, **we will rewrite this part for a general $V$**. The only reason to take $V=I_d$ is to allow for explicit computations in Proposition 1; **we will clarify this point.** Specifically, the modified drift in equation (8) will contain the term $P^{\perp}\_{Y_t^i} [V \mathsf{A}] dt$  instead of $P^{\perp}\_{Y_t^i}[\mathsf{A}]dt$.. For motivation on why training adds a deterministic component across successive layers, we kindly invite the reviewer to look at the works [4, 5].
>
> As for Assumption 2, $V=I_d$ is used in the same way **only to make the threshold explicit.** We will clarify this in the revision.
>
> ### Adressing Issue 2: Insights for ML Community and real Transformers
>
>
> We thank the reviewer and **we will add a Discussion section** detailing the practical implications of our findings. Specifically, we believe we believe our work brings two concrete insights to the ML community:
>
> - **Clustering / Rank Collapse (Th. 2):** We mathematically prove that injecting randomness (even though it may be considered as a step to prevent rank collapse) is **insufficient to prevent rank collapse**, although it alters the clustering geometry.
>
> Since **rank collapse in real pretrained transformers is already a well-documented empirical phenomenon** [2, 6], our work delivers a valid theoretical mechanism explaining why this behavior persists despite both stochasticity and residual connection.
>
> - **Initialization Design (Th. 1):** Our result proves that the **limiting behavior of deep transformers depends only on the first two moments of the weights.** We believe this contributes to the discussion on the choice of random initializations (Gaussian, uniform, etc.).
>
>
> ### Addressing Q1: $Q$ and $K$ Matrices
> In Section 4, we assume $Q$ and $K$ are constant matrices shared across layers **with no additional assumptions** on their structure. While our analytical method does generalize to certain classes of random and independent $Q$ and $K$ at each layer, the resulting limit becomes **significantly** less interpretable  and heavily dependent on the distribution of $K^\top Q$. Because of this complexity, we leave the fully random $Q$ and $K$ setting for future work.
>
> Meanwhile **importance of adding randomness through the matrix $V$ is empirically validated** on СIFAR-10 in Section 6.
>
> ### Addressing Q2: Intuition Behind Theorem 1
> The spherical constraints imposed by **RMS normalization force the tokens to move along the surface of a sphere, meaning their "velocity" vectors must be strictly tangent to the sphere**. In the deterministic case, this yields the dynamic $dY_t=P^\perp_{Y_t}[V\mathsf{A}(Y_t, \textbf{Y}_t)]\,dt$. We show in the paper that substituting $V\,dt$ with Itô noise in the stochastic case naturally maintains this orthogonal projection.
>
> Beyond this, we will add more comments and intuition to the mathematical results in the revised version.
>
>
> ### References
>
> [1] Karagodin et al. Clustering in causal attention masking. NeurIPS 2024.
>
> [2] Geshkovski et al. The emergence of clusters in self-attention dynamics. NeurIPS 2023.
>
> [3] Sander et al. Sinkformers: Transformers with doubly stochastic attention. AISTATS 2022.
>
> [4] Cohen et al. Scaling properties of deep residual networks. ICML 2021.
>
> [5] Marion et al. Scaling ResNets in the large-depth regime. JMLR 2025.
>
> [6] Dong et al. Attention is not all you need: Pure attention loses rank doubly exponentially with depth. ICML 2021.

---

> > ### Author Rebuttal · Reviewer_24Mv · 2026-04-04
> >
> > Thanks for the authors' response. I appreciate the clarifications they have provided. However, my concerns about the paper's practical relevance have not been fully addressed. I still find the model rather toy-like, and some of its assumptions (for example, treating the KV as a constant matrix) appear to be introduced primarily for the sake of making the theory tractable, rather than as natural or well-motivated simplifications. For this reason, I have decided to keep my score unchanged.

---

### Official Review · Reviewer_bEL5 · 2026-03-11

**Soundness:** 4
**Presentation:** 3
**Significance:** 4
**Originality:** 4
**Overall Recommendation:** 5
**Confidence:** 3

**Summary:**

This paper investigates the token dynamics in transformers with many layers, showing that once stochasticity is introduced tokens converge to antipodal points on the sphere, instead of a previous idea of them clustering to a common point.

Through a continuous time limit, the authors show that tokens obey a certain stochastic differential equation. They then use this result and introduce a normalization and noise parameter, epsilon, in order to study how noise influences the dynamics of tokens.

They are able to show a noise-induced phase transition from several perspectives: in the (d-beta) plane, showcasing the initiation of antipodal convergence, in terms of validation accuracy, and in terms of a transition in the (beta, epsilon) plane.
This can be done by introducing the attractors in probability space as order-parameters of the system and measure when this points become attractive.

This new phenomenon is supported by extensive simulations.

**Compliance With Llm Reviewing Policy:**

Affirmed.

**Key Questions For Authors:**

Please see points above.

**Limitations:**

The authors well-address the short-comings of their theory and mention explicitly these points as opportunities for future research.

**Strengths And Weaknesses:**

Soundness:

The submission is technically sound.
The authors provide rigorous proofs of their theorems and lemmas, and reference to other literature in certain cases.
A main result is that the token dynamics follow a certain stochastic differential equations when the token vector is trated in the continuous time limit.
The simulation results for the N = 2 case are in accordance with their mathematical description.
Of note is their simplification of assumptions in order to analytically track the N = 2 case, while also providing simulations for N > 2, reinforcing their work.

-------------------------------------------------------------------------

Presentation:

The submission is well structured and well written, with an easy to follow narrative.

The introductory remarks of the work position it relative to existing literature, especially in the treatment of stochastic differential equations on the sphere.

The authors discuss differences to previous literature, not so much about concurrent literature.

Of note is that the authors claim to have partially solved an existing problem of previous work, yet the reference given is most closely related to "Problem 3" instead of "Problem 2.15" in their reference:

"Geshkovski, B., Letrouit, C., Polyanskiy, Y., and Rigollet,
P. A mathematical perspective on transformers. Bulletin
of the American Mathematical Society, 62(3):427–479,
2025" .

I believe this to be a typo.

The link between projection onto the tangent space of a sphere and the RMS normalization in their fig. 1 is not sufficiently clear to the present reviewer.
The appendix can be confusing due to reliance on past results, more introductory sentences to where the used results come from and what they represent would be helpful.

-------------------------------------------------------------------------

Significance:

The paper offers a deeper understanding of clustering in deep Transformers, relevant to current machine learning research.
Often deeper understanding can lead to optimization of systems, which is so much needed as AI demand grows.

The authors claim to have partially solved an existing problem related to token dynamics with random Q,K,V matrices, having addressed the random V matrix case.
While the impact is specialized, it does link stochastics and partial differential equations to machine learning.
This issue is practially relevant, as their Fig.2 shows an impact of random weights on validation accuracy.

The N > 2 and the treatment of random Q,K cases are potential fruitful future directions for research, which these authors contribute in unlocking.

-------------------------------------------------------------------------

Originality:

The paper is grounded in pre-existing methods applied to a pre-existing system, but the combination of both is novel and deepen our understanding.

This work certainly highlights important properties of existing methods, as they use tools from stochastics and differential equations modeling, namely the introduction of noise epsilon, to demonstrate a phase transition.
The work demonstrates a new phenomenon: antipodal convergence of tokens.

The reason behind all assumptions is well articulated: namely the continuous time limit, introduction of noise, random initialization, normalization and simplifying assumptions.

The contributions are clearly distinguished from closely related works.

---

> ### Author Rebuttal · Authors · 2026-03-30
>
> Dear Reviewer,
>
> We sincerely thank you for your thoughtful review and for finding our work interesting. Below, we address your valuable comments in order:
>
> > *Typo in the reference Geshkovski, B., Letrouit, C., Polyanskiy, Y., and Rigollet, P. A mathematical perspective on transformers. Bulletin of the American Mathematical Society, 62(3):427–479, 2025*
>
> The discrepancy arises because the numbering differs between the arXiv preprint and the published BAMS version. **We will add a remark** in the revised text noting that this corresponds to Problem 3 in the preprint.
>
> > *The link between Projection / RMS and Figure 1 is not entirely clear.*
>
> We thank the reviewer for pointing this out; we will add a clearer explanation in the text. Our primary goal was to visually demonstrate that a **substantial component of the attention matrix $\mathsf{A}$** for tokens located on the sphere **is directed outwards, essentially driving an "explosion" or pushing the tokens away from the sphere**. RMS normalization, however, eliminates the component of $\mathsf{A}$ that is collinear to each token, preserving only the orthogonal component. It is precisely this **orthogonal part that forces the tokens to contract and cluster**.
>
> > *The appendix can also be difficult to follow due to reliance on previous results; additional introductory explanations clarifying the origin and role of these results would improve readability.*
>
> We will follow the reviewer’s suggestion and include additional context in the appendix to better explain the connection to previous results.

---

> > ### Author Rebuttal · Reviewer_bEL5 · 2026-04-03
> >
> > The authors have addressed all my points, which were minor remarks and I still support its acceptance.

---

### Official Review · Reviewer_CwwL · 2026-03-11

**Soundness:** 4
**Presentation:** 4
**Significance:** 3
**Originality:** 3
**Overall Recommendation:** 5
**Confidence:** 4

**Summary:**

This paper studies what happens to token representations in a simplified deep Transformer (attention-only, single head, no MLP) when the value matrices are drawn randomly and independently at each layer. Under a diffusion scaling of the residual connection, the paper proves that the discrete token dynamics converge in distribution to a system of SDEs on the sphere as the number of layers goes to infinity. For the case of two tokens, the paper shows that the randomness from initialization can enable antipodal configurations with positive probability via a phase transition governed by the attention temperature and token dimension, whereas in the deterministic setting tokens always collapse to a single point.

**Compliance With Llm Reviewing Policy:**

Affirmed.

**Final Justification:**

I maintain my score of 5. The paper would be a wonderful contribution to ICML in understanding the effect of noise in Transformers in the limit. The paper has a few weaknesses, but the authors have answered all of my questions related to these so I maintain my score recommendation.

**Key Questions For Authors:**

1. Assumption 1 requires bounded support ($|v^{kl}_n| \leq M$ a.s.), which excludes the untruncated Gaussian initialization used in the experiments (Section 6). Is the boundedness condition a strong technical requirement for the proof of Theorem 1 (e.g., for controlling remainder terms in the linearization step), or can it be relaxed to a sub-Gaussian tail condition? A positive answer here would strengthen my assessment of the paper's soundness.

2. The multi-token experiments (Figures 6, 7) track only whether at least one antipodal pair exists. A more informative diagnostic would be the effective rank of the $N \times d$ token matrix at the final layer, which directly measures representation diversity. Have the authors measured this quantity? If the effective rank remains $O(1)$ even with random $V$, the practical significance of the antipodal phenomenon would be substantially diminished. Conversely, if effective rank grows with $N$, that would significantly strengthen the paper's claims and would change my overall assessment.

3. The analysis relies on RMSNorm, which constrains tokens to the sphere $S^{d-1}$. Transformers also use LayerNorm, which subtracts the mean before normalizing, constraining tokens to the intersection of $S^{d-1}$ with a hyperplane. Do the authors expect the phase transition to hold under LayerNorm, and if so, would the boundary $\beta_c(d)$ shift?

**Limitations:**

The main limitation that the authors have not discussed is that the theory applies strictly at initialization before any training, which is mentioned only in passing in Section 4.2.

**Strengths And Weaknesses:**

## Strengths
- **Soundness.** The proofs are sound. The core strategy for Theorem 1 - linearization, Gronwall estimate, Watanabe's martingale problem, Slutsky transfer - is correctly applied. The moment identities (equation 18) that determine the SDE generator are correct, and Lemma 1's verification of sphere invariance via Ito's formula is correct. I found one potential error in the proof of Lemma 5: the scale function derivative should carry an exponent of $1/2$ on the prefactor (i.e., $[(1-x_0^2)/(1-x^2)]^{1/2}$, not $(1-x_0^2)/(1-x^2)$, due to a missing factor in $\int y/(1-y^2)\,dy$), but the asymptotics derived from this formula appear to be correct, so I think Theorem 2 and all downstream results are unaffected.

- **Originality.** The distinction between common noise (all tokens driven by the same matrix-valued Brownian motion $W_t$, arising from shared $V$ matrices) and idiosyncratic noise (independent Brownian motions per token, as in prior stochastic Transformer models) is genuine and well-articulated. This structural difference is what enables the antipodal phase transition.

- **Originality.** The limiting SDE (6) depends on the initialization distribution only through $\sigma^2$: any centered, bounded, fixed-variance entry distribution yields the same diffusion limit. This Donsker-type universality is a good technical contribution.

- **Presentation.** The threshold $\beta_c(d) = \frac{1}{2}\cosh^{-1}(d-2)$ in Theorem 2 is sharp and interpretable, and partially answers Problem 2.15 of Geshkovski et al. (2025).

## Weaknesses

- **Significance.** Theorem 2 establishes $P(A_1) + P(A_2) = 1$ for $N=2$: tokens must either coincide or become antipodal, with no intermediate angles. This means noise does not prevent rank collapse - it changes the collapse from a single point (rank 0) to a binary outcome where antipodal configurations (rank 1) occur with positive but often small probability (Figure 3 shows $P(A_2) \leq 0.30$). Going from effective rank 0 to rank at most 1 is a marginal improvement. The paper does not characterize the token geometry for large $N$, and does not discuss the specific technical obstructions that prevent extending the analysis beyond $N = 2$.

- **Significance.** All the results in the paper rely on the fact that for $N=2$, the inner product $Z_t = \langle Y^1_t, Y^2_t \rangle$ satisfies a self-contained scalar SDE, which admits a complete long-time analysis via classical one-dimensional diffusion theory. For $N \geq 3$, I don't know if this reduction is available, and the paper does not characterize the token geometry for large $N$ or discuss the specific technical obstructions that prevent extending the analysis.

- **Presentation** Theorem 1 is a statement about $L \to \infty$; Theorem 2 is about $t \to \infty$ in the limiting SDE. Combining them to make a meaningful conclusion requires the fact the limits can be exchanged, which is not justified in the paper.

---

> ### Author Rebuttal · Authors · 2026-03-30
>
> Dear Reviewer,
>
> We sincerely thank you for your careful reading of our work. We are happy to hear that you found our contributions interesting. We also thank you for pointing out the typos, which we will certainly correct in the next version of the paper. We are happy to address your questions and concerns:
>
> ### Addressing W1 and W2: Large $N$ Behavior
>
> We thank the reviewer for pointing out that we did not sufficiently comment on this aspect. In fact, the analysis of the large $N$ limiting behavior is challenging for several reasons:
>
> 1. As noted, it cannot be reduced to a 1D autonomous diffusion. For $N \geqslant 3$ (with constant $K, Q$), it requires studying an autonomous **system of** $\frac{N(N-1)}{2}$ **highly correlated 1D diffusions**.
> 2. Unlike the constant $V$ case, the stochastic **limiting dynamics do not constitute a gradient flow for the energy functional** $\mathsf{E}_{\beta},$ see [1], so the analysis does not reduce to studying energy extrema stability.
> 3. Particles are subject **only** to common noise, yielding a PDE with common noise rather than a standard McKean–Vlasov equation for $N \to \infty$.
>
> We will add a discussion regarding these technical obstructions for $N \geqslant 3$ to the revised version of the work.
>
> ### Addressing W3: Exchange of the Limits
>
> > Combining $L\to\infty$ and $t\to\infty$ to make a meaningful conclusion requires the fact the limits can be exchanged, which is not justified in the paper.
>
> We will clarify this in the appendix. In the standard topology on $\Omega = C(\mathbb{R}_{\geqslant 0}; (\mathbb{R}^d)^{N})$, sets $A_1$ and $A_2$ have the entire space as their boundary, preventing direct application of the Aleksandrov (*Portmanteau*) Theorem. Thus, probability calculations for continuous-time diffusions *do not automatically translate to networks with finite depth*; however, as confirmed by experiments (Fig. 2), they serve as a valid proxy. We note that transferring results from continuous limits back to discrete dynamics **is non-trivial even in deterministic settings** [1, 2] and requires delicate convergence estimates. We agree on the importance of such analysis and defer it to future work.
>
>
> ### Addressing Q1: Boundedness Assumption
>
> Even though boundedness is indeed used to estimate the remainder in the linearized dynamics, primarily **we inherited** this assumption from [3] (see Assumption IV), and it is a persistent technical assumption in the diffusion approximation literature (see [4] A1a).
>
> Nevertheless, we believe that the more general approach from [5, Section 7.5] could be adapted to our setting to help replace this requirement with a more natural **second-moment condition**. We will mention this in the updated version.
>
>
> ### Addressing Q2: Effective Rank and Token Diversity
>
> We sincerely thank the reviewer for pointing this out. **We will supplement Figures 6 and 7 in accordance with the reviewer's suggestion**: graphs showing the average rank of the system ([anonymous link to new figures](https://anonymous.4open.science/r/deep_stochastic_transformers-3FB3/rank_analysis.pdf)). This explicitly demonstrates that adding common noise **is insufficient to prevent rank collapse**, even as the number of tokens increases. We view this as a valuable negative result: adding realistic noise does not regularize away rank collapse, despite the a priori plausibility that it might. Our results therefore (i) bring the analysis closer to practical initializations, (ii) clarify that noise alone is insufficient to recover token diversity, and (iii) motivate working on explicit anti-collapse mechanisms, which we will mention as future work.
>
> ### Addressing Q3: LayerNorm and RMS
>
> The idea of studying LayerNorm is fruitful for future work. Preliminary experiments show that **applying LayerNorm on top of RMS  changes the threshold for antipodal configurations** ([anonymous link to preliminary experiments](https://anonymous.4open.science/r/deep_stochastic_transformers-3FB3/antipodal_fraction_modes.pdf)). We will mention this in the next version.
>
> ### Addressing the Limitation: Theory at Initialization
>
> > The main limitation that the authors have not discussed is that the theory applies strictly at initialization before any training, which is mentioned only in passing in Section 4.2.
>
> We agree and will move the remark about this to the very beginning of the paper.
>
> ### References
>
> [1] Geshkovski et al. A mathematical perspective on transformers. Bull. Amer. Math. Soc. 2025.
>
> [2] Geshkovski et al. The emergence of clusters in self-attention dynamics. NeurIPS 2023.
>
> [3] Watanabe. Diffusion approximations of some stochastic difference equations II. Hiroshima Math. J. 1984.
>
> [4] Kushner & Huang. On the weak convergence of general stochastic difference equations to a diffusion. SIAM J. Appl. Math. 1981.
>
> [5] Ethier & Kurtz. Markov Processes: Characterization and Convergence. 1986.

---

> > ### Author Rebuttal · Reviewer_CwwL · 2026-04-03
> >
> > I thank the authors for taking the time to address all the weaknesses and questions I raised. I'm happy to know that noise alone is insufficient to prevent rank collapse. I believe that's a big step toward understanding stochastic transformers. I truly appreciate the authors taking the time and effort to generate plots based on my question and in their interest in incorporating my suggestions.

---

### Official Review · Reviewer_b7GE · 2026-03-11

**Soundness:** 3
**Presentation:** 4
**Significance:** 3
**Originality:** 4
**Overall Recommendation:** 5
**Confidence:** 4

**Summary:**

The paper studies a simplified deep Transformer model in which each layer consists of self-attention with residual scaling $L^{-1/2}$, token-wise RMS normalization, and randomness entering through the value matrices. The main contribution is a diffusion-limit theorem showing convergence of the discrete dynamics to an interacting SDE on a product of spheres driven by common matrix-valued Brownian noise. Building on this limit, the paper analyzes long-time clustering behavior, and in the two-token case derives an explicit phase transition between single-cluster collapse and antipodal separation as a function of inverse temperature and ambient dimension. The paper also introduces a hybrid deterministic/stochastic model to isolate the effect of noise, and supplements the theory with numerical experiments, including simulations of the phase transition and a CIFAR-10 comparison.

**Compliance With Llm Reviewing Policy:**

Affirmed.

**Key Questions For Authors:**

* Which steps in the analysis fundamentally rely on freezing (Q) and (K)? Do the authors expect Theorem 1, or at least its qualitative conclusions, to survive when (Q) and (K) are also random across layers?

* What technical obstacles do the authors face when dropping the compact support assumption 1?

* In the regime (N>2), what is the conjectured long-time structure of the limiting dynamics? Is the intended picture a bipolar partition, repeated pair formation, or something else, and how should “clustering” be defined in that setting?

* Could the authors provide additional finite-depth evidence directly probing the predicted (1/L) versus (1/\sqrt{L}) scaling, and clarify whether the CIFAR-10 gain from random (V) should be interpreted as an optimization/training benefit, a representational effect, or both?

* Perhaps the authors could expand a bit more on whether they think that the stochastic behavior they showcase could be of interest in practice either for training or for generalization of these models, even if only at a at the speculative level? In other words, what insight can be gained from this work?

**Limitations:**

yes

**Strengths And Weaknesses:**

Strengths:
- The question is relevant and timely. Understanding what intrinsic initialization noise does to deep attention dynamics is a meaningful theoretical problem, and the paper takes a genuinely new step by deriving a stochastic limit from the architecture rather than imposing external token-wise noise by hand. In that sense, the modeling contribution is original.
- The main mathematical results are interesting and clearly presented. The diffusion limit under token-wise RMS normalization is nontrivial, and the two-token phase transition is a concrete and interpretable statement. The paper is generally well written, the results are clearly stated, and the proofs appear careful and, as far as I can see, technically sound. The only improvement I would suggest is that the probabilistic setup around natural filtrations/adaptedness could be clarified.
-  The numerical section is useful. The experiments support the predicted transition, suggest that antipodal behavior can persist beyond two tokens, and the CIFAR-10 comparison indicates that the stochastic initialization mechanism is not only a formal device but may have practical consequences.

Weaknesses:
The main limitations lie, in my view, in the paper's (sometimes implicit) assumptions:
- Randomness is injected only through (V), but (Q) and (K) are frozen in the theory, even though they determine the interaction geometry through the attention weights. Since the core nonlinearity sits in the attention kernel, leaving (Q) and (K) deterministic means that the paper does not yet address one of the main stochastic features of standard transformer initialization.
- The strongest clustering/synchronization statement is limited to (N=2), and the explicit phase boundary further relies on Assumption 2, (Q^\top K = I_d). The multi-token regime is treated only numerically.
- The compact support assumption on the weights' initialization distribution, ruling out for instance Gaussian initialization, appears somewhat artificial.

Overall, I find the paper original, well executed, and clearly presented. The theoretical analysis is challenging and some assumptions have to be made, but I still find the paper provides an interesting insight and some nontrivial and original results.

---

> ### Author Rebuttal · Authors · 2026-03-30
>
> Dear Reviewer,
>
> We sincerely thank you for your careful reading and valuable feedback.
>
> As suggested (''*the probabilistic setup around natural filtrations/adaptedness could be clarified*''), we will certainly expand the introductory section in the Appendix and add the requested details regarding adaptivity and filtrations.
>
> Other Questions and Weakesses related to them, we will address one by one:
>
> ### Addressing Q1 and W1: Constant $K$ and $Q$
>
> We agree with the reviewer that analyzing random $K$ and $Q$ is a critically important next step.
>
> - **What steps rely on freezing $Q$ and $K$?** The **averaging step** in the proof of Theorem 1, as well as **representing the limiting diffusion as an SDE** system.
>
> - **Do the qualitative conclusions survive?** Yes, the method actually generalizes **without changes** for some random $Q$ and $K$ (e.g., when $K^{\top}Q$ has i.i.d. uniform entries), but the characterization we obtain is **significantly** less interpretable. We leave a full study of random $Q$ and $K$ for future work.
>
> Furthermore, unlike with $V$, **the limiting process distribution for random $Q$ and $K$ is not solely determined by the first two moments**. This means that for different $K^{\top}Q$ initializations, the diffusion properties might change significantly.
>
> **We will add a remark in the revised text** clarifying exactly how random $Q$ and $K$ impact the limiting dynamics and why we leave this complex extension for future work.
>
> ### Addressing Q2 and W3: Bounded Support Assumption
>
> Primarily, **we adopted the compact support assumption from the diffusion approximation literature** [1], which explicitly requires the $1/\sqrt{L}$ and $1/L$ increments to be bounded with probability $1$ on any ball in $X$. Secondly, it is technically required to control the remainder terms during linearization.
>
> We agree with the reviewer that this assumption seems rather redundant. Although it is persistent in the diffusion approximation literature (see A1a in [2]), we believe that the more general approach from [3, Section 7.5] could be adapted to our setting to replace this requirement with a **more natural second-moment condition**. We plan to leverage these results in future work and will mention this possibility in the updated version.
>
> ### Addressing Q3 and W2: Large $N$ Behavior
>
> For the multiple token ($N>2$) case, we numerically observe a significant rank reduction to $0$ or $1$ (*meaning the tokens collapse to either a single point or two antipodal points*). This confirms our expectation that rank collapse, which is how we define "*clustering*" in this setting, does indeed occur.
>
> Overall, we conjecture that for a broad class of both random and deterministic $Q$ and $K$, the **set of stable configurations derived for two tokens will not expand in the $N>2$ regime**. A detailed analysis of exactly which configurations are stable, and crucially, **the rate of this convergence** (that can be different from case to case), is left for future work.
>
> ### Addressing Q4: Diffusion Scaling
>
> While a comprehensive and detailed **analysis of scaling** in transformers could serve as a topic for a separate work (taking into account practically used initializations and normalizations), more general studies on residual networks (which supersum Transformers) show that the $1/\sqrt{L}$ scaling is important for both optimization/training (*gradient stability*) and representation (*feature diversity*) (see [4, 5] and references therein).
>
> Our choice of the diffusion scaling in the CIFAR-10 experiments was primarily motivated by these insights.
>
>
> ### Addressing Q5: Practical Insights
>
> First, our Donsker-type result establishes a universality principle for initialization, proving that limiting dynamics depend only on the first two moments of the weights and making the specific choice of distribution (Gaussian, uniform, etc.) irrelevant at large depth.
>
>
> Second, our findings serve as a constructive negative result. While stochastic initialization slightly diversifies the clustering picture (e.g., enabling antipodal configurations), the core attention architecture still inevitably induces rank collapse. **Noise in $V$ alone cannot prevent this.** However, it could change the rate at which collapse occurs. Figure 9(a,b) provides a preliminary observation to this effect; however, a detailed analysis of these rates is left for future work.
>
>
> ### References:
>
> [1] Watanabe. Diffusion approximations of some stochastic difference equations II. Hiroshima Math. J. 1984.
>
> [2] Kushner & Huang. On the weak convergence of general stochastic difference equations to a diffusion. SIAM J. Appl. Math. 1981.
>
> [3] Ethier & Kurtz. Markov Processes: Characterization and Convergence. 1986.
>
> [4] Marion et al. Scaling ResNets in the large-depth regime. JMLR. 2025.
>
> [5] Yang et al. Tensor Programs VI: Feature Learning in Infinite Depth Neural Networks. ICLR. 2024.

---

> > ### Author Rebuttal · Reviewer_b7GE · 2026-04-03
> >
> > The authors responded to most of my questions. Many of thos replies deferred the questions to future work. I will keep my score unchanged.

---

### Decision · Program_Chairs · 2026-04-30

**Decision:**

Accept (spotlight)

**Comment:**

This paper analyzes token dynamics in deep transformers with random initialisation. The authors take the limiting interacting-particle system point of view, and prove a Donsker-type convergence theorem for these dynamics. All four reviewers found the work original, mathematically rigorous, and clearly presented, with three recommending acceptance and one giving a weak reject. The positive reviews highlight the novel treatment of common noise, the interpretable phase transition, and the careful numerical validation. While one reviewer criticised the theoretical assumptions, this AC found them to be in line with the current literature, to which this paper makes a valuable contribution. I therefore recommend acceptance.